



# First long-term and near real-time measurement of atmospheric trace elements in Shanghai, China

Yunhua Chang[1, 2], Kan Huang[3], Congrui Deng[3], Zhong Zou[4], Shoudong Liu[1, 2], and Yanlin Zhang[1, 2 *]

[1]Yale-NUIST Center on Atmospheric Environment, International Joint Laboratory on Climate and Environment Change (ILCEC), Nanjing University of Information Science & Technology, Nanjing 210044, China

[2]Key Laboratory of Meteorological Disaster, Ministry of Education (KLME)/ Collaborative Innovation Center on Forecast and Evaluation of Meteorological

Disasters (CIC-FEMD), Nanjing University of Information Science & Technology, Nanjing 210044, China

[3]Center for Atmospheric Chemistry Study, Shanghai Key Laboratory of Atmospheric Particle Pollution and Prevention (LAP[3]), Department of Environmental Science and Engineering, Fudan University, Shanghai 200433, China

[4]Pudong New Area Environmental Monitoring Station, Shanghai 200135, China

*Correspondence to*: Yanlin Zhang (dryanlinzhang@outlook.com or zhangyanlin@nuist.edu.cn)

**Abstract**: Atmospheric trace elements, especially metal species, are an emerging environmental and health concern with poorly constrained on its abundances and

sources in Shanghai, the most important industrial megacity in China. Here we continuously performed a one-year (from March 2016 to February 2017) and hourly-resolved measurement of eighteen elements in fine particles ($PM_{2.5}$) at Shanghai urban center with a Xact multi-metals monitor and several collocated instruments. Independent ICP-MS offline analysis of filter samples was used to validate the

performance of Xact that was based on energy-dispersive X-ray fluorescence analysis of aerosol deposits on reactive filter tapes. Mass concentrations (mean±1σ; ng m$^{-3}$) determined by Xact ranged from detection limits (nominally 0.1 to 20 ng m$^{-3}$) to 14.7 μg m$^{-3}$, with Si as the most abundant element (638.7±1004.5), followed by Fe





(406.2±385.2), K (388.6±326.4), Ca (191.5±383.2), Zn (120.3±131.4), Mn (31.7±38.7),

Pb (27.2±26.1), Ba (24.2±25.4), V (13.4±14.5), Cu (12.0±11.4), Cd (9.6±3.9), As (6.6±6.6), Ni (6.0±5.4), Cr (4.5±6.1), Ag (3.9±2.6), Se (2.6±2.9), Hg (2.2±1.7), and Au (2.2±3.4). Metal related oxidized species comprised an appreciable fraction of $PM_{2.5}$ during all seasons, accounting for 8.3% on average. As a comparison, atmospheric metal pollution level in Shanghai was comparable with other industrialized cities in

East Asia but one or two orders magnitude higher than the sites in North America and Europe. Here our high time-resolution observations over long-term period also offer a unique opportunity to provide robust diurnal profiles for each species, which are useful in determining the sources and processes contributing to the fluctuation of atmospheric trace elements. Besides, various mathematical methods and physical evidences were

served as criteria to constrain various solutions of source identification. Results showed that atmospheric trace elements pollution in Shanghai was the interplay of local emissions and regional transport, and different sources of metal species generally have different variation patterns associated with different source regions. Specifically, V and Ni were confirmed as the prominent and exclusive tracer of heavy oil combustion from

shipping traffic. Fe and Ba were strongly related to brake wear, and exhibited significant correlation with Si and Ca, suggesting that Si and Ca in Shanghai were primarily sourced from road fugitive dust rather than long-distance dust transport and local building construction sites. Stationary combustion of coal was found to be the major source of As, Se, Pb, Cu and K, and the ratio of As/Se was used to infer that coal

consumed in Shanghai likely originated from Henan coal fields in Northern China. Cr, Mn and Zn were the mixed result of emissions from stationary combustion coal, ferrous metals production, and nonferrous metals processing. Ag and Cd in Shanghai urban atmosphere were also the mixture of miscellaneous sources. Collectively, our findings in this study provide baseline data with high detail, which are needed for developing

effective control strategies to reduce the high risk of acute exposure to atmospheric trace elements in China's megacities.



## 1. Introduction

It has been broadly recognized that personal exposure to atmospheric aerosols have detrimental consequences and aggravating effects on human health such as respiratory, cardiovascular, and allergic disorders (Pope III et al., 2002; Pope III et al., 2009; Shah et al., 2013; West et al., 2016; Burnett et al., 2014). Among the chemical components relevant for aerosol health effects, airborne heavy metals (a very imprecise term without authoritative definition (Duffus John, 2002), loosely refers to elements with atomic density greater than 4.5 g cm$^{-3}$ (Streit, 1991)) are of particular concern as they typically feature with unique properties of bioavailability and bioaccumulation (Morman and Plumlee, 2013; Tchounwou et al., 2012; Fergusson, 1990; Kastury et al., 2017), representing 7 of the 30 hazardous air pollutants identified by the US Environmental Protection Agency (EPA) in terms of posing the greatest potential health threat in urban areas (see www.epa.gov/urban-air-toxics/urban-air-toxic-pollutants). Depending on aerosol composition, extent and time of exposure, previous studies have confirmed that most metal components of fine particles (PM$_{2.5}$; particulate matter with aerodynamic diameter equal to or less than 2.5 μm) exerted a multitude of significant diseases from pulmonary inflammation, to increased heart rate variability, to decreased immune response (Fergusson, 1990; Morman and Plumlee, 2013; Leung et al., 2008; Hu et al., 2012; Pardo et al., 2015; Kim et al., 2016).

Guidelines for atmospheric concentration limits of many trace metals are provided by the World Health Organization (WHO) (WHO, 2005). In urban atmospheres, ambient trace metals typically represent a small fraction of PM$_{2.5}$ on a mass basis, while metal species like Cd, As, Co, Cr, Ni, Pb and Se are considered as human carcinogens even in trace amounts (Iyengar and Woittiez, 1988; Wang et al., 2006; Olujimi et al., 2015). It has been shown that Cu, Cr, Fe and V have several oxidation states that can participate in many atmospheric redox reactions (Litter, 1999; Brandt and van Eldik, 1995; Seigneur and Constantinou, 1995; Rubasinghege et al., 2010a), which can catalyze the generation of reactive oxygenated species (ROS) that have been associated with direct



molecular damage and with the induction of biochemical synthesis pathways (Charrier and Anastasio, 2012; Strak et al., 2012; Rubasinghege et al., 2010b; Saffari et al., 2014; Verma et al., 2010; Jomova and Valko, 2011). Additionally, lighter elements such as Si, Al and Ca are the most abundant crustal elements next to oxygen, which can typically

constitute up to 50% of elemental species in remote continental aerosols (Usher et al., 2003; Ridley et al., 2016). These species are usually associated with the impacts of aerosols on respiratory diseases and climate (Usher et al., 2003; Tang et al., 2017).

Health effects of airborne metal species are not only seen from chronic exposure, but also from short-term acute concentration spikes in ambient air (Kloog et al., 2013;

Strickland et al., 2016; Huang et al., 2012). In addition, atmospheric emissions, transport, and exposure of trace metals to human receptors may depend upon rapidly evolving meteorological conditions and facility operations (Tchounwou et al., 2012; Holden et al., 2016). Historically accepted ambient trace metals sampling devices generally collect 12 to 24-hr integrated average samples, which are then sent off to be

lab analyzed in a time-consuming and labor-intensive way. As a consequence, daily integrated samples inevitably ignore environmental shifts with rapid temporality, and thereby hinder the efforts to obtain accurate source apportionment results such as short-term metal pollution spikes related to local emission sources. In fact, during a short-term trace metals exposure event, 12 or 24-hr averaged sample concentrations for metal

species like Pb and As may be orders of magnitude lower than the 4-hr or 15-min average concentration from the same day (Cooper et al., 2010). Current source apportionment studies are mainly performed by statistical multivariate analysis such as receptor models (e.g., Positive Matrix Factorization, PMF), which could greatly benefit from high inter-sample variability in the source contributions through increasing the

sampling time resolution. In this regard, continuous monitoring of ambient metal species on a real-time scale is essential in the trace metals sources and their health assessment studies.

Currently, there are only a few devices available for the field sampling of ambient



aerosols with sub-hourly or hourly resolution, i.e., the Streaker sampler, the DRUM
(Davis Rotating-drum Unit for Monitoring) sampler, and the SEAS (Semi-continuous
Elements in Aerosol Sampler) (Visser et al., 2015b; Visser et al., 2015a; Bukowiecki et
al., 2005; Chen et al., 2016). Mass loadings of trace metals collected by these samplers
can be analyzed with high sensitive accelerator-based analytical techniques, in
particular particle-induced X-ray emission (PIXE) or synchrotron radiation X-ray
fluorescence (SR-XRF) (Richard et al., 2010; Bukowiecki et al., 2005; Maenhaut, 2015;
Traversi et al., 2014). However, a major drawback of the analysis is that they require a
large commitment of analytical time. More recently, aerosol time-of-flight mass
spectrometry (Murphy et al., 1998; Gross et al., 2000; DeCarlo et al., 2006), National
Institute for Standards and Technology (NIST)-traceable reference aerosol generating
method (QAG) (Yanca et al., 2006), distance-based detection motif (Cate et al., 2015),
environmental magnetic properties coupled with support vector machine (Li et al.,
2017), and Xact$^{TM}$ 625 automated multi-metals analyzer (Fang et al., 2015; Jeong et al.,
2016; Phillips-Smith et al., 2017; Cooper et al., 2010) have been developed for more
precise, accurate, and frequent measurement of ambient metal species. The Xact
method is based on nondestructive XRF analysis of aerosol deposits on reactive filter
tapes, which has been validated by US Environmental Technology Verification testing
and several other field campaigns (Fang et al., 2015; Phillips-Smith et al., 2017; Jeong
et al., 2016; Yanca et al., 2006; Cooper et al., 2010).

Located at the heart of the Yangtze River Delta, Shanghai is home to nearly 25 million
people as of 2015, marked as the largest megacity in China (Chang et al., 2016).
Shanghai city is one of the main industrial centers of China, playing a vital role in the
nation's heavy industries, including but not limited to, steel making, petrochemical
engineering, thermal power generation, auto manufacture, aircraft production, and
modern shipbuilding (Normile, 2008; Chang et al., 2016; Huang et al., 2011). Besides,
Shanghai is China's most important gateway for foreign trade, which has the world's
busiest port, handled over 37 million standard containers in 2016 (see
www.simic.net.cn/news_show.php?lan=en&id=192101). As a consequence, Shanghai



is potentially subject to substantial quantities of trace metal emissions (Duan and Tan, 2013; Tian et al., 2015). Ambient concentrations of trace metals, especially Pb and Hg,

in Shanghai atmospheres have been sporadically reported during the past two decades (Shu et al., 2001; Lu et al., 2008; Wang et al., 2013; Zheng et al., 2004; Huang et al., 2013; Wang et al., 2016). Of current interest are V and Ni, which are often indicative of heavy oil combustion from ocean-going vessels (Fan et al., 2016; Liu et al., 2017). Still, previous work rarely illustrated a full spectrum of metal species in ambient

aerosols. Meanwhile, most available source evidences were inferred based on filter sampling and off-line analysis, which were not necessarily representative of actual origins. Furthermore, recent attribution of hospital emergency-room visits in China to $PM_{2.5}$ constituents failed to take short-term variations of trace metals into account (Qiao et al., 2014), which could inevitably underestimate the toxicity of aerosols and

potentially misestimate the largest influence of aerosol components on the health effects (Honda et al., 2017).

In this study, the first of its kind, we conducted a long-term and near real-time measurement of atmospheric trace metals in $PM_{2.5}$ with a Xact multi-metals analyzer in Shanghai, China, from March 2016 to February 2017. The primary target of the present

study is to elucidate the atmospheric abundances, variation patterns and source contributions of trace elements under complex urban environment, which can be used to support future health studies.

## 2. Methods

### 2.1 Field measurements

#### 2.1.1 Site description

Figure 1a shows the map of eastern China with provincial borders and land covers, in which Shanghai city (provincial level) sits in the middle portion of China's eastern coast and its metropolitan area (indicated as densely-populated area in Fig. 1b) concentrated on the south edge of the mouth of the Yangtze River. The municipality borders the



provinces of Jiangsu and Zhejiang to the north, south and west, and is bounded to the

east by the East China Sea (Fig. 1a). Shanghai has a humid subtropical climate and

experiences four distinct seasons. Winters are chilly and damp, with northwesterly

winds from Siberia can cause nighttime temperatures to drop below freezing. Air

pollution in Shanghai is low compared to other cities in northern China like Beijing,

but still substantial by world standards, especially in winter (Han et al., 2015; Chang et

al., 2017).

Field measurements were performed at the rooftop (~18 m above ground level) of

Pudong Environmental Monitoring Center (PEMC; 121.5447°E, 31.2331°N; ~7 m

above sea level) in Pudong New Area of southwestern Shanghai with dense population

(Fig. 1b). Pudong New Area is described as the "showpiece" of modern China due to

its height-obsessed skyline and export-oriented economy. For PEMC, there were no

metal-related sources (excepting for on-road traffic) and high-rise buildings nearby to

obstruct observations, and air mass could flow smoothly. More broadly, as indicted in

Fig. 1c, PEMC is surrounded by multitudinous emissions sources like coal-fired power

plants (CFPP) in its four directions and iron and steel smelting in the northwest. Besides,

a large amount of ship exhaust emissions in 2010 such V (Fig. 1d) and Ni (Fig. 1e) in

the YRD and the East China Sea within 400 km of China's coastline was recently

quantified based on an automatic identification system model (Fan et al., 2016).

Therefore, PEMC can be regarded as an ideal urban receptor site of various emission

sources. More information regarding the sampling site has been given elsewhere

(Chang et al., 2017; Chang et al., 2016).

**2.1.2 Hourly elemental species measurements**

From March 1$^{st}$ 2016 to February 28$^{th}$ 2017, hourly ambient mass concentrations of

eighteen elements (Si, Fe, K, Ca, Zn, Mn, Pb, Ba, V, Cu, Cd, As, Ni, Cr, Ag, Se, Hg,

and Au) in PM$_{2.5}$ were determined by a Xact multi-metals monitor (Model Xact$^{TM}$ 625,

Cooper Environmental Services LLT, OR, USA) (Phillips-Smith et al., 2017; Jeong et

al., 2016; Fang et al., 2015; Yanca et al., 2006). Specifically, the Xact sampled the air





on a reel-to-reel Teflon filter tape through a $PM_{2.5}$ cyclone inlet (Model VSCC-A, BGI
Inc., MA, USA) at a flow rate of 16.7 L min$^{-1}$. The resulted $PM_{2.5}$ deposit on the tape

was automatically advanced into the analysis area for nondestructive energy-dispersive
X-ray fluorescence analysis to determine the mass of selected elemental species as the
next sampling was being initiated on a fresh tape spot. Sampling and analysis were
performed continuously and simultaneously, except during advancement of the tape
(~20 sec) and during daily automated quality assurance checks. For every event of

sample analysis, the Xact included a measurement of pure Pd as an internal standard to
automatically adjust the detector energy gain. XRF response was calibrated using thin
film standards for each metal element of interest. These standards were provided by the
manufacturer of Xact, producing by depositing vapor phase elements on blank
Nuclepore (Micromatter Co., Arlington, WA, USA). The Nuclepore filter of known area

was weighed before and after the vapor deposition process to determine the
concentration ($\mu$g cm$^{-2}$) of each element. In this study, excellent agreement between the
measured and standard masses for each element was observed, indicating a deviation
of < 5%. The 1-hr time resolution minimum detection limits (in ng m$^{-3}$) were: Si (17.80),
K (1.17), Ca (0.30), V (0.12), Cr (0.12), Mn (0.14), Fe (0.17), Ni (0.10), Cu (0.27), Zn

(0.23), As (0.11), Se (0.14), Ag (1.90), Cd (2.50), Au (0.23), Ba (0.39), Hg (0.12), and
Pb (0.13).

As a reference method to validate Xact on-line measurements, daily $PM_{2.5}$ samples were
also collected at PEMC site using a four-channel aerosol sampler (Tianhong, Wuhan,
China) on 47 mm cellulose acetate and glass filters at a flow rate of 16.7 L min$^{-1}$. The

sampler was operated once a week with a 24-hr sampling time (starting from 10:00 am).
In total 48 filter samples (21 cellulose acetate filter samples and 27 glass filter samples)
were collected, in which 8 paired samples were simultaneously collected by cellulose
acetate and glass filters. In the laboratory, elemental analysis procedures were strictly
followed the latest national standard method "Ambient air and stationary source

emission-Determination of metals in ambient particulate matter-Inductively coupled
plasma/mass spectrometry (ICP-MS)" (HJ 657-2013) issued by the Chinese Ministry



of Environmental Protection. A total of 24 elements (Al, Fe, Mn, Mg, Mo, Ti, Sc, Na, Ba, Sr, Sb, Ca, Co, Ni, Cu, Ge, Pb, P, K, Zn, Cd, V, S, and As) were measured using an Inductively coupled plasma-mass spectrometry (ICP-MS; Agilent, CA, USA). The results of the 8 paired samples were compared firstly. Significant correlations were observed for species of K, Cr, Mn, Fe, Ni, Cu, As, Cd, Ba, Zn, and Pb (Table S1), and these species were used to validate the performance of Xact. In Table S1, the slope values of Cr (1.9) and Ba (2.6) were higher than other species, this can be explained by higher background values of Cr and Ba collected by cellulose acetate filters.

**2.1.3 Auxiliary measurements, quality assurance and quality control**

Hourly mass concentrations of $PM_{2.5}$ were measured using a Thermo Fisher Scientific TEOM 1405-D. Data on hourly concentrations of CO, $NO_2$, and $SO_2$ were provided by PEMC. Meteorological data, including ambient temperature ($T$), relative humidity (RH), wind direction (WD) and wind speed (WS), were provided by Shanghai Meteorological Bureau at Century Park station (located approximately 2 km away from PEMC). All the above online measurement results were averaged to a 1 hr resolution.

As data in the current study were collected in near-real time, the importance for quality assurance and quality control (QA/QC) system can be crucial in order to improve data quality throughput. The routine procedures, including the daily zero/standard calibration, span and range check, station environmental control, and staff certification, were followed the Technical Guideline of Automatic Stations of Ambient Air Quality in Shanghai based on the national specification HJ/T193–2005, which was modified from the technical guidance established by the USEPA. QA/QC for the Xact measurements was implemented throughout the campaign. The internal Pd, Cr, Pb, and Cd upscale values were recorded after the instrument's daily programmed test, and the $PM_{10}$ and $PM_{2.5}$ cyclones were cleaned weekly.

**2.2 Data analysis**

**2.2.1 Statistical analysis**



To identify possible sources of measured trace metals, three methods of statistical

analysis, i.e., correlation matrix, principle component analysis (PCA), and hierarchical

clustering of reordering correlation matrix, were performed. The "corrplot" package in

R is a graphical display of a correlation matrix, confidence interval. It also contains

specific algorithms to do matrix. More information regarding corrplot can be found at

CRAN.R-project.org/package=corrplot (Wei and Simko, 2016).

Spearman correlations were firstly performed to establish correlations between trace

metals, which can be used to investigate the dependence among multiple metal species

at the same time. The result is a table containing the correlation coefficients between

each variable and the distribution of each metal species on the diagonal. Secondly,

principle component analysis (PCA) with a varimax rotation (SPSS Statistics® 24,

IBM®, Chicago, IL, USA) was performed on the measured data set, which have been

used widely in receptor modelling to identify major source categories. The technique

operates on sample-to-sample fluctuations of the normalized concentrations. It does not

directly yield concentrations of species from various sources but identifies a minimum

number of common factors for which the variance often accounts for most of the

variance of species (e.g., (Venter et al., 2017), and references therein). The trace metal

concentrations determined for the 18 species were subjected to multivariate analysis of

Box-Cox transformation and varimax rotation, followed by subsequent PCA. Lastly,

we applied agglomeration strategy for hierarchical clustering, a method of cluster

analysis which seeks to build a hierarchy of clusters, to mining the hidden structure and

pattern in the correlation matrix (Murdoch and Chow, 1996; Friendly, 2002). In order

to decide which clusters should be combined as a source, or where a cluster should be

split, a measure of dissimilarity between sets of observations is required, Ward's method

is served as a criterion applied in hierarchical cluster analysis. The "corrplot" package

can draw rectangles around the chart of correlation matrix (indicated as a potential

source) based on the results of hierarchical clustering.

## 2.2.2 Conditional probability function and bivariate polar plot for tracing source





**regions**

The determination of the geographical origins of trace metals in Shanghai requires the use of diagnostic tools such as conditional probability function (CPF) and bivariate

polar plot (BPP), which are very useful in terms of quickly gaining an idea of source impacts from various wind directions and have already been successfully applied to various atmospheric pollutants and pollution sources (Chang et al., 2017; Carslaw and Ropkins, 2012). In this study, the CPF and BPP were performed on the one-year data set for the major trace metals with similar source. The two methods have been made in

the R "openair" package and are freely available at www.openair-project.org (Carslaw and Ropkins, 2012).

The CPF is defined as CPF = $m_\theta/n_\theta$, where $m_\theta$ is the number of samples in the wind sector $\theta$ with mass concentrations greater than a predetermined threshold criterion, and $n_\theta$ is the total number of samples in the same wind sector. CPF analysis is capable to

show which wind directions are dominated by high concentrations and give the probability of doing so. In this study, 90[th] percentile of a given metal species was set as threshold, and 24 wind sectors were used ($\Delta\theta = 15^0$). Calm wind ($< 1$ m s$^{-1}$) periods were excluded from this analysis due to the isotropic behavior of wind vane under calm winds.

The BPP demonstrate how the concentration of a targeted species varies synergistically with wind direction and wind speed in polar coordinates, which thus is essentially a non-parametric wind regression model to alternatively display pollution roses but include some additional enhancements. These enhancements include: plots are shown as a continuous surface and surfaces are calculated through modelling using smoothing

techniques. These plots are not entirely new as others have considered the joint wind speed-direction dependence of concentrations (see for example Liu et al. (2015)). However, plotting the data in polar coordinates and for the purposes of source identification is new. The BPP has described in more detail in Carslaw et al. (2006) and the construction of BPP had been presented in our previous work (Chang et al., 2017).





**3 Results and discussion**

**3.1 Mass concentrations**

**3.1.1 Data overview and comparison**

The temporal patterns and summary statistics of hourly elemental species
concentrations determined by the Xact at PEMC during March 2016-Feburary 2017 are

reported in Fig. 2. The mass concentrations of 18 elements measured in Shanghai were
sorted from high to low in Fig. 3. The one-year data set presented in the current study,
to the best of our knowledge, represents the longest on-line continuous measurement
series of atmospheric trace metals.

Taking the study period as a whole, ambient average mass concentrations of elemental

species varied between detection limit (ranging from 0.05 to 20 ng m$^{-3}$) and nearly 15
µg m$^{-3}$, with Si as the most abundant element (mean ± 1σ; 638.7 ± 1004.5 ng m$^{-3}$),
followed by Fe (406.2 ± 385.2 ng m$^{-3}$), K (388.6 ± 326.4 ng m$^{-3}$), Ca (191.5 ± 383.2)
ng m$^{-3}$, Zn (120.3 ± 131.4 ng m$^{-3}$), Mn (31.7 ± 38.7 ng m$^{-3}$), Pb (27.2 ± 26.1 ng m$^{-3}$),
Ba (24.2 ± 25.4 ng m$^{-3}$), V (13.4 ± 14.5 ng m$^{-3}$), Cu (12.0 ± 11.4 ng m$^{-3}$), Cd (9.6 ± 3.9

ng m$^{-3}$), As (6.6 ± 6.6 ng m$^{-3}$), Ni (6.0 ± 5.4 ng m$^{-3}$), Cr (4.5 ± 6.1 ng m$^{-3}$), Ag (3.9 ±
2.6 ng m$^{-3}$), Se (2.6 ± 2.9 ng m$^{-3}$), Hg (2.2 ± 1.7 ng m$^{-3}$), and Au (2.2 ± 3.4 ng m$^{-3}$).
According to the ambient air quality standards of China (GB 3095-2012), EU
(DIRECTIVE 2004/107/EC) and WHO, the atmospheric concentration limits for Cd,
Hg, As, Cr (VI), Mn, V, and Ni are 5, 50 (1000 for WHO), 6 (6.6 for WHO), 0.025, 150

(WHO), 1000 (WHO), and 20 (25 for WHO) ng m$^{-3}$, respectively. Therefore, airborne
metals pollution in Shanghai is generally low by the current limit ceilings. Nevertheless,
information regarding the specific metal compounds or chemical forms is rarely
available given that most analytical techniques only record data on total metal content.
In the absence of this type of information, it is generally assumed that many of the

elements of anthropogenic origin (especially from combustion sources) are present in
the atmosphere as oxides. Here we reconstructed the average mass concentrations of



metal and crustal oxides as 5.2, 5.0, 2.8, and 3.1 μg m$^{-3}$ in spring, summer, fall, and winter, respectively, with the annual average concentration was 3.9 μg m$^{-3}$, which accounting for 8.3% of total PM$_{2.5}$ mass (47 μg m$^{-3}$) in 2016. Detailed calculation of the reconstructed mass has been fully described elsewhere (Dabek-Zlotorzynska et al., 2011).

The toxicological effect of hazardous metal species is more evident and well aware in soils and aquatic ecosystems, while few (if any) studies on the geochemical cycle of trace metals have considered the fast dynamics of trace metals in the atmosphere. Using a diversity of chemical, physical, and optical techniques, elevated atmospheric concentrations of various metal species have been observed globally; however, a tiny minority of them were performed in a way of high time-resolved. As a comparison, we compiled previous work related to the near real-time measurements of trace metals concentrations in Table 1. The concentrations of most trace metals in Shanghai were commonly an order or two orders of magnitude higher than the sites measured in Europe and North America, while generally ranged in the same level as industrialized city like Kwangju in South Korea. Exceptionally, the concentrations of V and Ni in Shanghai were up to three times higher than that of Kwangju City. This can be expected since Shanghai has the world's busiest container port, and V and Ni were substantially and almost exclusively emitted from heavy oil combustion in ship engines of ocean-going vessels (see more discussion in Section 3.2 and 3.3).

### 3.1.2 Variations at multiple time scales

In contrast to traditional trace metals measurements, the on-line XRF used in the current study enables measurement of metal species concentrations with 1 hr resolution, which are useful both for source discrimination and in determining the processes contributing to elevated trace metals levels through investigation of their diurnal cycles. We will also discuss weekly cycles because certain emission sources may make a pause or reduction during weekends. Additionally, Shanghai has a humid subtropical climate and experiences four distinct seasons, which could potentially exert an influence on the





mass concentrations of atmospheric metal species. Therefore, monthly and seasonal variations of ambient concentrations for each metal species were demonstrated. The variations of weather data, including $T$, RH, wind speed, wind direction, and precipitation, at PEMC during our study period were also illustrated in Fig. S1.

As depicted in Fig. 4, the seasonal-averaged mass concentrations of Ag and Cd stayed

exceptionally constant regardless of the season. Moreover, variations of Ag and Cd in Fig. 5 are highly consistent at weekly and diurnal basis, both exhibiting a sharp increase of (normalized) mass concentrations after midnight and then an outright decline after 1:00 (local time), and keeping quite steady during the rest of the day. Globally, anthropogenic emissions of Ag and Cd exceed the natural rates by well over an order

of magnitude. Thus, our results strongly signaled that Ag and Cd pollution in highly populous Shanghai had very stable and climate-independent anthropogenic emission sources.

Except for Ag and Cd, other metal species were seasonally variable without a uniform variation pattern (Fig. 4). Specifically, the concentrations of five metal species, i.e., As,

Cu, Hg, K, and Pb, were higher in winter and lower in summer, which were consistent with the seasonal variation pattern of aerosol organics, sulphate, nitrate, and ammonium that were fully-explored in China. Generally, severer air pollution in Eastern China during winter were mainly attributed to the accumulation of pollutants emitted from coal-based heating in conjunction with stagnant meteorological conditions (Huang et

al., 2013). However, more trace metals like Ba, Cr, Fe, Mn, Ni, V, and Zn shown the highest concentrations in spring, suggesting more complex sources and different physical/chemical formations of atmospheric metal species in Shanghai (see discussion later). A better example is Ca and Si, which presented the highest degree of seasonal variations with higher concentration levels in summer.

Like Ag and Cd, another example of covariation is V and Ni (Fig. 6). Diurnally, both V and Ni peaked at around 6:00 and then gradually decreased to the bottom at 14:00, which were generally in accord with wind speed (Fig. 6c), suggesting that V- and Ni-



containing aerosols in Shanghai undergo mid- to long-range atmospheric transport. Co-emitted from heavy oil combustion, previous studies in Shanghai port have concluded

that a ratio of 3.4 for V/Ni in ambient aerosols could be a reliable indicator of ship traffic source. Here performed in urban area, the average ratio of V/Ni in our study was 3.1 with slightly seasonal changes (Fig. 7), which was very close to the ratio of averaged V and Ni content in ship heavy fuel oils, indicating V- and Ni-containing aerosols subject to minor atmospheric transformation and thus can be served as a robust tracer

of shipping emissions even in costal urban areas. The office hours for Shanghai customs is Monday through Saturday, and the most important holiday in the Chinese Calendar-Lunar New Year-is usually celebrated during February. This can be used to explain that the weekly (monthly) lowest impact of shipping emissions in Shanghai was occurred during Sunday (Fig. 6c) (February (Fig. 6d)).

Distinctive diurnal variation patterns are observed for Si, Ca, Fe, Ba (Fig. 8) as well as Mn and Zn (Fig. 9), characterizing by two marked peaks at noon (10:00-11:00) and evening (17:00-18:00), and agreeing well with the diurnal variation of Shanghai traffic flow. Indeed, as the largest megacity in terms of economy and population in China, the contribution of vehicles, both exhaust and non-exhaust emissions (Thorpe and Harrison,

2008), to ambient trace metals cannot to be belittled in Shanghai. As demonstrated in Fig. 8d and Fig. 9d, there is an evident drop in the concentrations of Si, Ca, Fe, Ba, Mn and Zn after entering weekends. Data collected from Shanghai Traffic Administration Bureau, Fig. S2 shows seasonally average weekly cycles of on-road traffic flow in Shanghai, in which the average traffic flows in weekends were lower than that in

weekdays. Given that Fe is the support material for brake pad, and the agents present in brake linings typically consist of Zn, Mn and Ba, less traffic flow in weekends not only lower road suspend dust but also cut metal species emissions from brake wear (Zhao et al., 2015; Xie et al., 2008). Nevertheless, monthly variation pattern of Ca was different from other metals during April through July (Fig. 10c), and Si exhibited

unusually high levels in July (Fig. 8c) and 1:00 (Fig. 8b). PMF analysis of elemental species has suggested that Fe and Ba in urban atmospheres were primarily caused by



vehicular brake wear while high levels of Si and Ca were more likely driven by road resuspended dust (Heo et al., 2009; Harrison et al., 2012; Jeong et al., 2016; Amato et al., 2011; Xie et al., 2008). Besides, construction activities are always thriving in China, which have been confirmed as an important source of Si- and Ca-rich crustal matters (Tian et al., 2015). Building construction activities in Shanghai are concentrated between the end of spring festival (normally the end of February) and the approach of midsummer heat (August) (Tian et al., 2015). For an annual mean of 16.1 °C, Shanghai averages 28.3 °C in August, and the municipal authority will impose a mandatory moratorium toward outdoor construction once temperature rises to 37 °C (Chang et al., 2016). Besides, heavy-duty diesel vehicles (mostly for transforming and dumping construction waste) in Shanghai can only be allowed to operate after midnight (Chang et al., 2016), leading to some higher emissions from diesel engines and unpaved roads.

In Fig. 10 and Fig. 11, diurnal variations of trace metals like Cu, K, Se, As, Pb, Au, and Hg are seemingly full of clutter. Interestingly however, the monthly and weekly cycles of the above-mentioned metal species are remarkably consistent, hinting at the possibility of sharing similar sources. Apart from anthropogenic activity, the planetary boundary layer height in Shanghai normally reaches its annual climax during August and September (Chang et al., 2016), which are favorable to the vertical dispersion of air pollutants, leading to the lowest concentration levels in Fig. 10. From a perspective of man-made emissions, coal combustion of industrial boilers and nonferrous metal smelting represent the dominant sources for Se/As/Pb/Cr, and Au/Hg, respectively. For Cu, China annually emits around 10000 metric tons Cu, which have long been thought to be primarily sourced from automotive braking because copper or brass is a major ingredient in friction material (Tian et al., 2015). However, our results based on field measurements in Shanghai go against the inherent notion of Cu origins. Gathering evidences have shown that topsoil and coal ash are also enriched in K (Schlosser, et al., 2002; Thompson and Argent, 1999; Westberg et al., 2003; Reff et al., 2009). These findings suggest more diverse sources of trace metals in a highly industrialized megacity like Shanghai.

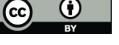

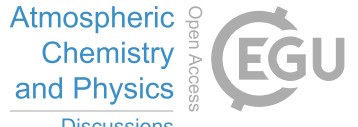Atmospheric
### 3.2 Source analysis

The goal of source analysis in the current study is twofold. On the one hand, we will use various mathematical and physical criteria to constrain various solutions of source apportionment. On the other hand, we will take CPF and BPP as diagnostic tools for quickly gaining the idea of potential source regions, which in turn will contribute to further analysis of source apportionment.

### 3.2.1 Pinpoint the best possible source

As the first approach in the source analysis, Spearman correlation matrixes were prepared for all measured elemental species and presented in Fig. 12. From Fig. 12, relatively good correlations are observed between trace metals associated with heavy oil combustion (i.e., V and Ni), brake and tire wear (e.g., Fe, Mn, Ba and Zn), trace crustal matters (i.e., Si and Ca), and coal combustion (e.g., Se, As and Pb), indicating the complex influence of multiple sources in Shanghai. Still, trace crustal matters are also well correlated with metal species related to brake and tire wear. As discussed in Section 3.1, the bimodal variations of Si and Ca are also evident, which jointly suggest that the trace crustal matters we observed in Shanghai urban atmosphere were primarily derived from road fugitive dust. Nevertheless, with approximately 70% of most of the airborne crustal species being present in the coarse size fraction (Huang et al., 2013), here we call for more size-resolved sampling and analysis of PM to provide a paramount view of atmospheric trace metals in the future.

Using multiple lines of evidence above, we've inferred heavy oil combustion from ship engines as the possible source of V and Ni. Here in Fig. 12, different from all other inter-correlated trace metals, V and Ni are not only highly correlated but also exclusively correlated to each other. For metal species data greater than their 90th percentile, CPF analysis in Fig. 13 shows that over 90% Ni and almost all V observed in our receptor site come from the west (East China Sea). From a perspective of geographical origin, Fig. 13 also clearly shows major air masses containing both V and



Ni are sourced from the southwest. Collectively, it can be safely concluded that shipping emissions are best possible source of V and Ni in Shanghai. The successful identification of the best possible source, i.e., shipping emissions, is helpful in terms of examining the feasibility of various source apportionment solutions. In other words, V and Ni should be clustered as a single factor/source in any solution.

### 3.2.2 Explore and constrain more sources

To determine sources of more trace metals, PCA was applied as an exploratory tool, since much larger datasets are required for definitive source apportionment with PCA. Therefore, only the most apparent groupings of metal species relating to expected sources in the region were identified. The factor loadings are presented in Fig. 14, indicating four statistically significant factors with eigenvalues equal to or greater than 1. These four factors obtained explained nearly 80% of the variance. Unexpectedly, PCA of the 18 elemental species did not reveal any meaningful factors. This can be preliminarily tested by the large contribution of V and Ni in two different factors, but more importantly, was attributed to the large influence of nonferrous smelting, coal combustion, and traffic-related emissions on ambient trace metals measured at PEMC (see discussion later).

Matrix reorder in conjunction with clustering analysis can be a powerful tool for mining the hidden structure and pattern in the correlation matrix of Fig. 12. Here six solutions of source apportionment, from 3 sources/factors to 8 sources/sources, determined by clustering analysis were presented in Fig. 15. As mentioned above, we can rule out the solution of 3 and 4 sources/factors in Fig. 15a and 15b at first since V and Ni are combined with Au and Hg as a single source.

In Fig. 15c through 15f, Au and Hg are always clustered together as an individual factor, suggesting certain similar source(s) they may share. Globally, anthropogenic sources, including a large number of industrial point sources, annually account for 2320 Mg of Hg released to the atmosphere (Nriagu, 1996). Despite varied in emission quantity



among different studies, Hg emissions from nonferrous metals production, coal consumption by industrial boilers and coal combustion by power plants incontrovertibly represent the top three anthropogenic sources in China, with a share of 33.1, 25.4 and 20.7% for each sector according to Tian et al. (2015). In other words, stationary combustion of coal contributes roughly half Hg emissions of anthropogenic

origins. The distribution of metal-related factories is illustrated in Fig. 1, in which coal-fired power plants encompass our site at every major direction while nonferrous metal-related works are concentrated in the west (especially in the northwest) but absent in the east of PEMC. Here we calculate percentile concentration levels of Au and Hg, and plots them by wind direction in Fig. 16a and 16b, respectively. It is clearly shows Au

and Hg are primarily derived from stationary combustion of coal because the overwhelmingly prevailing air masses from the west of PEMC. Even though, there are few 99-99.9[th] percentile data that are originated from the northwest of PEMC (Fig. 16a and 16b), signaling that nonferrous metals production only affect high percentile concentrations of Au and Hg at PEMC.

Statistically, the relatively weak correlation between Ag and Cd (Fig. 12) denies the possibility that they could equally share the same source. Therefore, the solution of 5 factors/sources illustrated in Fig. 15c is superficially invalid. Knowledge gap remains regarding the emissions of Ag in China, while this is not the case for Cd. Totally 456 metric tons in 2010, the major category sources for Cd emissions in China were

nonferrous metals smelting (mainly primary Cu smelting industry), coal consumption by industrial boilers and other non-coal sources, which accounted for 44.0, 22.8 and 8.4% of the total emissions, respectively (Tian et al., 2015). Significantly high concentrations of Ag were also observed near the field of nonferrous metals processing works and coal-fired power plants in China, suggesting that nonferrous metals

production and industrial coal consumption are also important source of Ag. As reported in Fig. 16c and 16d, the receptor site, PEMC, evenly receives air masses containing Ag and Cd in all directions, indicating that both Ag and Cd in Shanghai urban atmosphere are the mixture of miscellaneous sources.





Among the reaming metal species, Cu/K/Pb/As/Se, Cr/Mn/Zn, Fe/Ba, and Ca/Si are
grouped together as a population from first to last in Fig. 15. In China, 73% of As, 62%
Se, 56% of Pb, and 50% Cu emissions were found to be coal combustion (Tian et al.,
2015). Therefore, As and Se are typically treated as the specific tracers of coal-related
emissions in China (Tian et al., 2015; Zhang, 2010; Ren, 2006). Broadly, different coal
field has starkly different contents of As and Se, which offers opportunity to infer the
possible region of coal production. For example, the average As content in coal of
northern China (0.4-10 mg kg$^{-1}$) and southern China (0.5-25 mg kg$^{-1}$) are 3 mg kg$^{-1}$ and
10 mg kg$^{-1}$, respectively (Zhang, 2010; Ren, 2006). With an average value of 2 mg kg$^{-1}$,
Se content in various Chinese coal fields ranges from 0.1 mg kg$^{-1}$ to 13 mg kg$^{-1}$
(Zhang, 2010; Ren, 2006). We have no direct information regarding As and Se contents
in coal that used in Shanghai. As an alternative, here the average ratio of ambient As
and Se (As/Se ratio) in Shanghai was calculated as 2.65, which was very close to the
As/Se ratio (2.76) in coal (long flame coal of Jurassic period) of Henan Province in
northern China (Zhang, 2010). Note that the As/Se ratios in Shanxi Province, a major
coal production region in China, are ranged from is 0.24 in Taiyuan (the capital city of
Shanxi) to 1.96 in Datong (a prefecture-level city in northern Shanxi) (Zhang, 2010).
The emission source regions of Cu, K, Pb, As, and Se are demonstrated in Fig. 17, in
which all species are generally transported from far western Shanghai (where stationary
combustion of coal located; Fig. 1) to our sampling site.

It is well known that Ca and Si are two of the five most abundant elements in the Earth's
crust, enjoying a long reputation of being the most specific tracer of wind-blown dust.
Located on the eastern coast of China, Shanghai rarely receives long-range transport of
crustal matters from aeolian dust and Gobi deserts in northwestern China (Huang et al.,
2013). Sampling in the east side of Shanghai, a majority of measured Ca and Si must
be sourced from road fugitive dust or urban construction sites, which can be further
validated by the CPF and BPP analysis of Ca and Si (Fig. S3). As discussed above,
ambient Fe and Ba at PEMC are strongly associated with brake wear (Zhao et al., 2015).
In Fig. 15, significant correlations are observed among Ca, Si, Fe, and Ba, suggesting



that the measured Ca and Si during our study period were more likely derived from road fugitive dust.

As a group of tightly-linked metal species, identifying a principle source for measured Cr, Mn and Zn at PEMC is a daunting task. Cr, Mn and Zn have relatively good correlation with two groups of metal species, i.e., coal combustion-related emissions (i.e., Cu, K, Pb, As, and Se) and brake wear-related emissions (i.e., Fe and Ba). This is basically attributed to the diverse sources of Cr, Mn and Zn themselves in China. For

Cr, the national atmospheric emissions from anthropogenic sources in 2010 reached 7465.2 metric tons, of which about 5317.6 metric tons were emitted from coal consumption by industrial boilers (Tian et al., 2015). Coal combustion is also the dominant source of Mn in a national dimension, while in urban areas with intensive road network, the contribution of brake and tire wear to ambient Mn is appreciable.

Different from Cr and Mn, ferrous metals smelting sector (31.3%), especially the pig iron and steel production industry, is the leading contributor to Zn emissions in China, followed by coal combustion (21.7%) and nonferrous metals smelting sector (19.3%). The Baosteel company in Shanghai, located in the northwest of PEMC (approximately 20 km apart; Fig. 1), is a flagship manufacturer in terms of producing the first-chop iron

and steel throughout China. Fig. S4 also evidently reflect high concentrations of Zn can be occurred in the northwest of PEMC. In sum, ambient Cr, Mn and Zn in Shanghai urban atmosphere is highly mixed with coal combustion by industry sectors and power plants, ferrous metals production, and nonferrous metals smelting.

**4. Conclusion and outlook**

This paper presents the results from a year-long, near real-time measurement study of 18 trace elements (Si, Fe, K, Ca, Zn, Mn, Pb, Ba, V, Cu, Cd, As, Ni, Cr, Ag, Se, Hg, and Au) in $PM_{2.5}$ using a Xact multi-metal monitor, conducted at an urban site in Shanghai from March 2016 to February 2017. The scientific significance of this work can be reflected by the general findings as follows:



-The Xact multi-metals monitor was demonstrated as a valuable and practical tool for ambient monitoring of atmospheric trace elements through comparing online monitoring results with ICP analyses of offline filter samples.

-The metal concentrations in Shanghai are one or two orders of magnitude higher than in north America and Europe, highlighting the need to allocate more scientific, technical,

and legal resources on controlling metals' emissions in China.

-The total of metal related species comprised approximately 8.3% of the $PM_{2.5}$ mass, which should not be ignored in China's recent epidemiologic study of attributing hospital emergency-room visits to $PM_{2.5}$ chemical constituents.

-The full coverage of trace elemental species (18) measurement and the high temporal

frequency (hourly) in the work provided unprecedented details regarding the temporal evolution of metal pollution and its potential sources in Shanghai.

-Various mathematical methods (e.g., correlation matrix, hierarchical clustering, and conditional probability function) and physical evidences were used to infer the contribution of local emissions (specific sectors) and long-range transport to measured

metal species.

A greater value and more interesting topic to the scientific community would be to fully assess the role of $PM_{2.5}$ chemical constituents (including metal species) and sources of emission to human health. Looking into the future, three major moves will be taken toward thoroughly addressing these questions. Firstly, characterizing the chemical and

isotopic (including metal species) signatures of emission sources will be intensively taken through field samplings as well as laboratory simulations (see example of Geagea et al. (2007)). Secondly, Xact multi-metals monitor, Sunset OC/EC analyzer (Chang et al., 2017), and MARGA (Monitoring of AeRosols and Gases) platform will be collocated across a rural-urban-background transect to simultaneous measurement of

hourly metal species, carbonaceous aerosols, and inorganic aerosol components in $PM_{2.5}$. Lastly, integrating all available information regarding $PM_{2.5}$ chemical species



and isotopes into receptor model or atmospheric chemical transport model to create more specific and confident source apportionment results.

**Competing interests**

The authors declare that they have no competing interests.

**Data availability**

Data are available from the corresponding authors on request.

**Acknowledgements**

This study was supported by the National Key Research and Development Program of

China (2017YFC0210101), Provincial Science Foundation of Jiangsu (Grant no. BK20170946), University Science Research Project of Jiangsu Province (17KJB170011), National Science Foundation of  China (Grant nos. 91644103, 41603104, 41429501, and 91644105). Yunhua Chang and Zhong Zou acknowledge the support of the Start-up Foundation for Introducing Talent to NUIST and Shanghai

Pudong New Area Sci-tech Development Funds (Grant no. PKJ2016-C01), respectively. We also acknowledge the Qingyue Open Environmental Data Centre (http://data.epmap.org) for the unconditional help in terms of providing criteria pollutants monitoring data.

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



**Table 1.** A collection of long-term and high-time resolution measurements of ambient trace elements concentrations (ng m$^{-3}$) in fine particles.

| Species | Shanghai, CN[a] | Wood Buffalo, CA[b] | Gwangju, KP[c] | London, UK[d] | London, UK[e] | Barcelona, ES[f] | Toronto, CA[g] |
|---|---|---|---|---|---|---|---|
| Ag | 3.9 | / | / | / | / | / | / |
| As | 6.6 | / | 9.6 | / | / | / | / |
| Au | 2.2 | / | / | / | / | / | / |
| Ba | 24.2 | / | 52.0 | 10.3 | 3.7 | / | 1.9 |
| Ca | 191.5 | 54.0 | 122 | 78.7 | 50.1 | 130.0 | 54.0 |
| Cd | 9.6 | / | / | / | / | / | / |
| Cr | 4.5 | 0.04 | / | 2.3 | 0.8 | 8.0 | 0.24 |
| Cu | 12.0 | 2.04 | 15.5 | 12.8 | 4.9 | 8.0 | 3.1 |
| Fe | 406.2 | 60.0 | 293.0 | 350.3 | 118.9 | 131.0 | 76.8 |
| Hg | 2.2 | / | / | / | / | / | / |
| K | 388.6 | 31.0 | 732.0 | 27.2 | 23.7 | 82.0 | 27.1 |
| Mn | 31.7 | 1.12 | 24.0 | 4.8 | 2.5 | 6.0 | 1.8 |
| Ni | 6.0 | 0.08 | 3.8 | 0.5 | 0.2 | 3.0 | 0.21 |
| Pb | 27.2 | / | 49.0 | 2.3 | 1.8 | 12.0 | 2.4 |
| Se | 2.6 | / | 4.3 | / | / | / | 0.3 |
| Si | 638.7 | 143 | / | / | / | / | / |
| V | 13.4 | 0.21 | 4.6 | 1.3 | 0.6 | 8.0 | 0.11 |
| Zn | 120.3 | 0.88 | 103.0 | 8.9 | 5.3 | 25.0 | 11.3 |

**Note**: a, this study; b, Phillips-Smith et al., 2017; c, Park et al., 2014; d, PM$_{0.3-2.5}$, Marylebone Road (Visser et al., 2015b); e, PM$_{0.3-2.5}$, North Kensington (Visser et al., 2015b); f, Road site (Dall'Osto et al., 2013); g, Sofowote et al., 2015. We noticed that although a huge data set of hourly resolved trace metals had been reported in Jeong et al. (2016) and Visser et al. (2015a), in which no detailed information regarding the specific mass concentrations of trace metals were given.





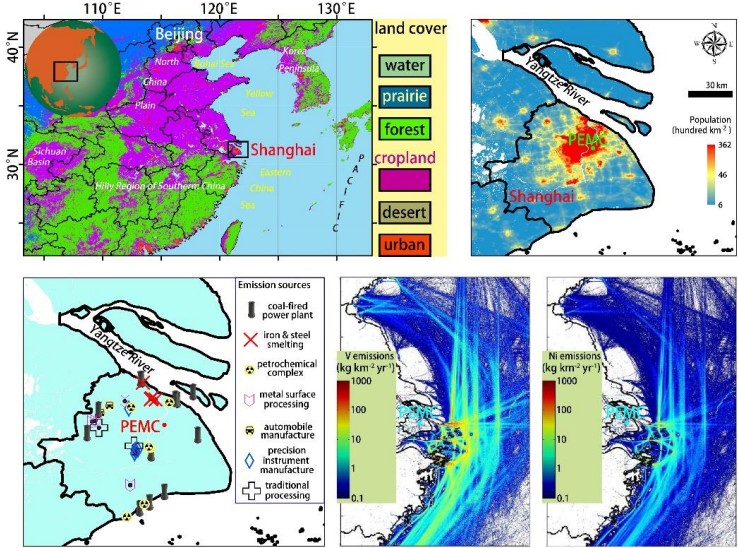

**Figure 1.** A land uses map indicating the location of Shanghai (**a**; black box), as well
as the population density (**b**) and the major point sources (**c**) around the sampling site
(PEMC). The emissions of V (**d**) and Ni (**e**) from shipping in the YRD and the East
China Sea within 400 km of the coastline were estimated based on an automatic
identification system model (adopted from (Fan et al., 2016)).






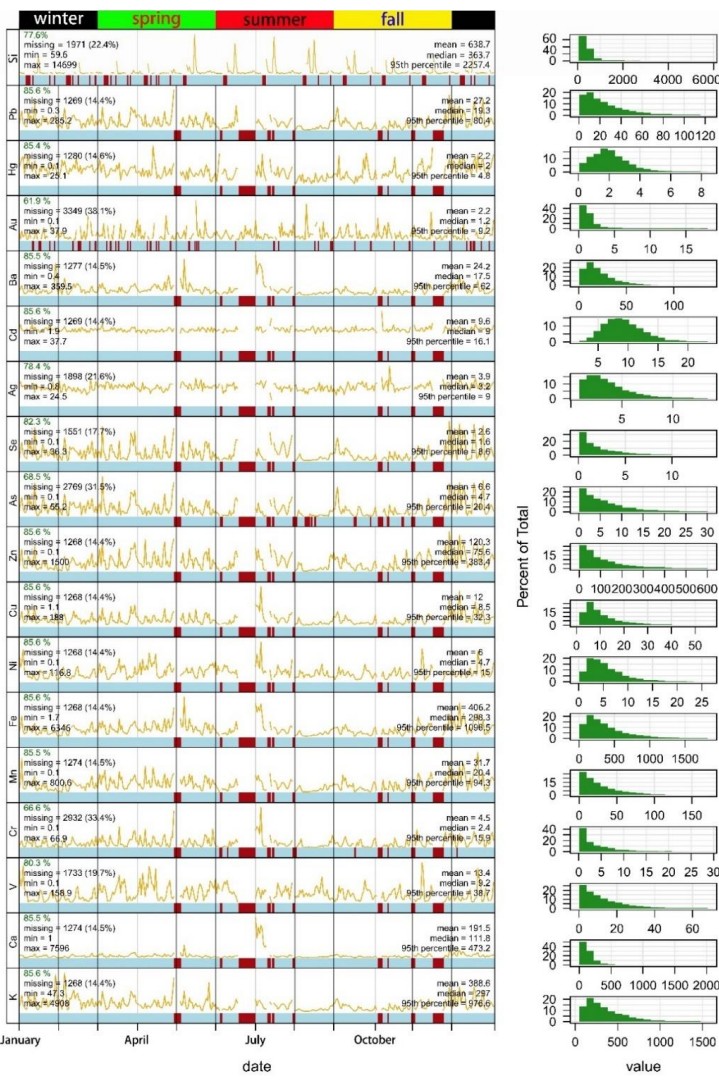

**Figure 2.** General statistical summaries of 18 trace elements measured in Shanghai. The plots in the left panel show the time series data, where blue shows the presence of data and red shows missing data. The mean daily values are also shown in pale yellow scaled to cover the range in the data from zero to the maximum daily value. As such, the daily values are indicative of an overall trend rather than conveying quantitative information. For each elemental species, the overall summary statistics are given. The panels on the right show the distribution of each elemental species using a histogram plot.





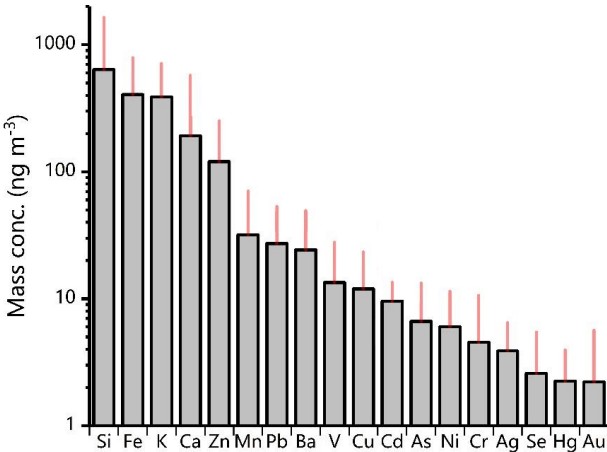

**Figure 3.** A quick glance of the mass concentrations of 18 trace elements measured in Shanghai as sorted from high to low (log10 scaling). The dark red line indicates one standard deviation.











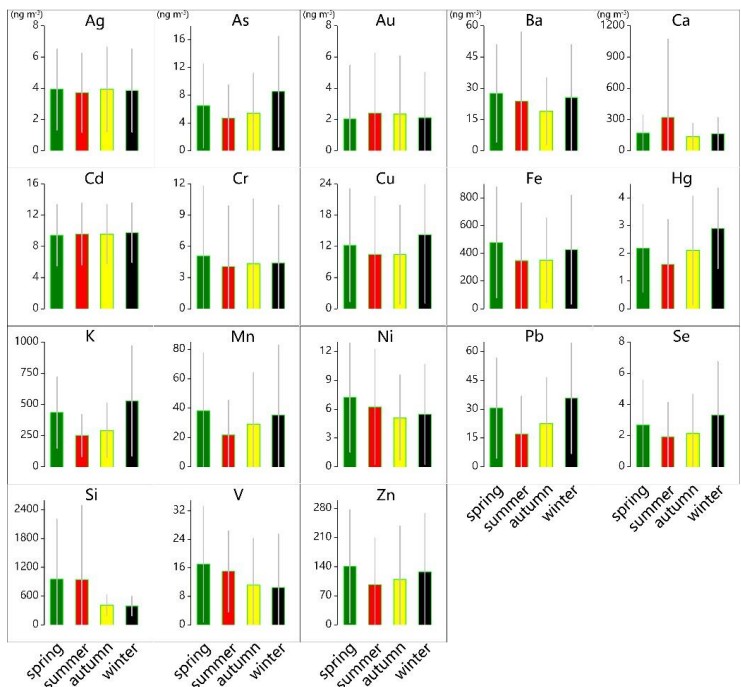

**Figure 4.** Seasonal variations of mass concentrations for 18 trace elements measured in Shanghai between March 2016 and February 2017. The gray line indicates one two standard deviations. Four seasons in Shanghai were defined as follows: March-May as spring, June-August as summer, September- November as fall, and December and January-February as winter.




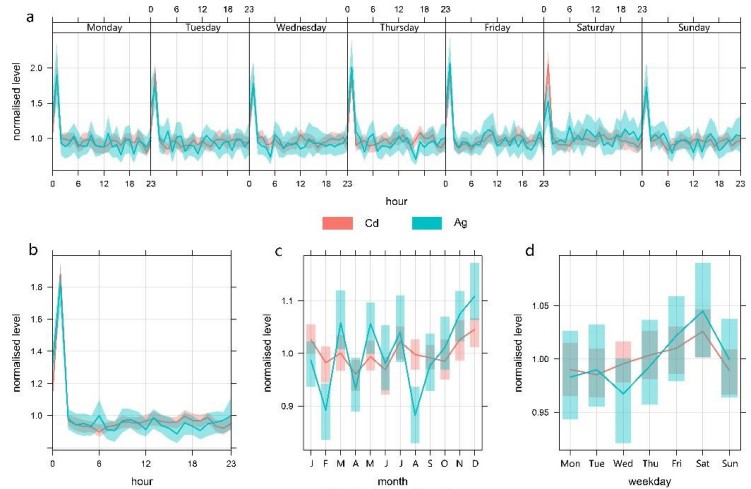

**Figure 5.** Weekly diurnal (**a**), diurnal (**b**), monthly (**c**), and weekly (**d**) variations of normalized Cd and Ag concentrations in Shanghai.





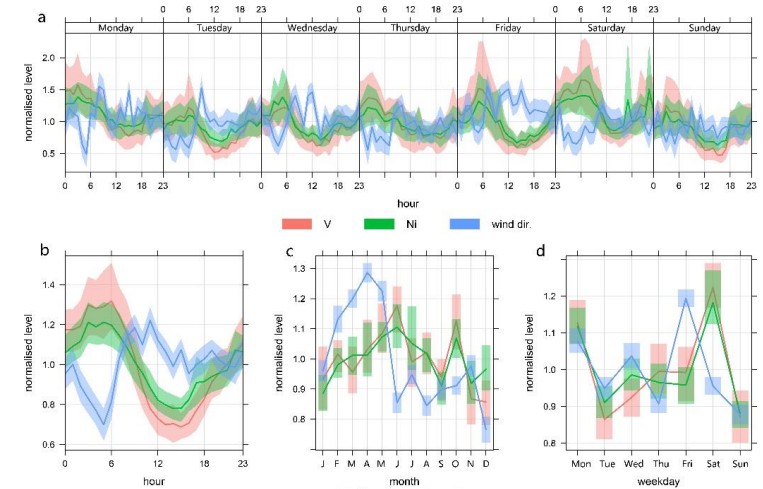


**Figure 6.** Weekly diurnal (**a**), diurnal (**b**), monthly (**c**), and weekly (**d**) variations of normalized V and Ni concentrations, and wind directions in Shanghai.







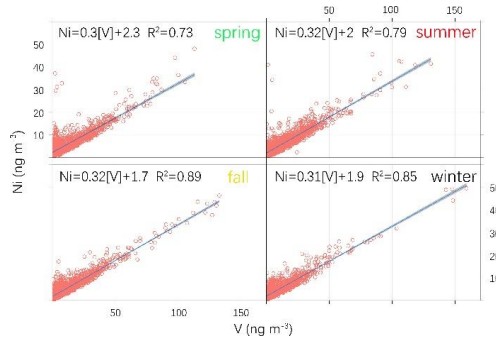

**Figure 7.** Linear correlation analysis between V (x axis) and Ni (y axis) in Shanghai
during four seasons.








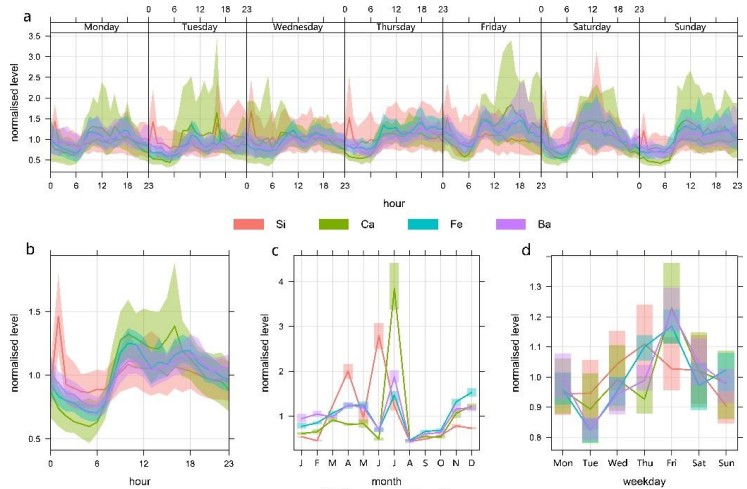

**Figure 8.** Weekly diurnal (**a**), diurnal (**b**), monthly (**c**), and weekly (**d**) variations of normalized Si, Ca, Fe, and Ba concentrations in Shanghai.

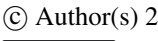



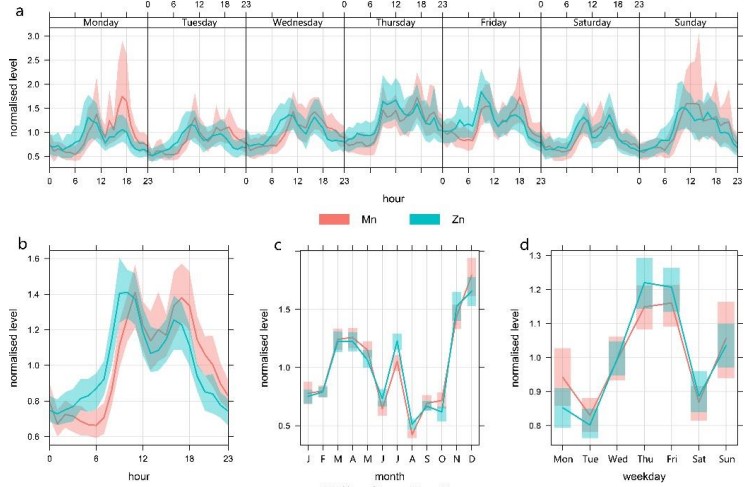

**Figure 9.** Weekly diurnal (**a**), diurnal (**b**), monthly (**c**), and weekly (**d**) variations of normalized Mn and Zn concentrations in Shanghai.





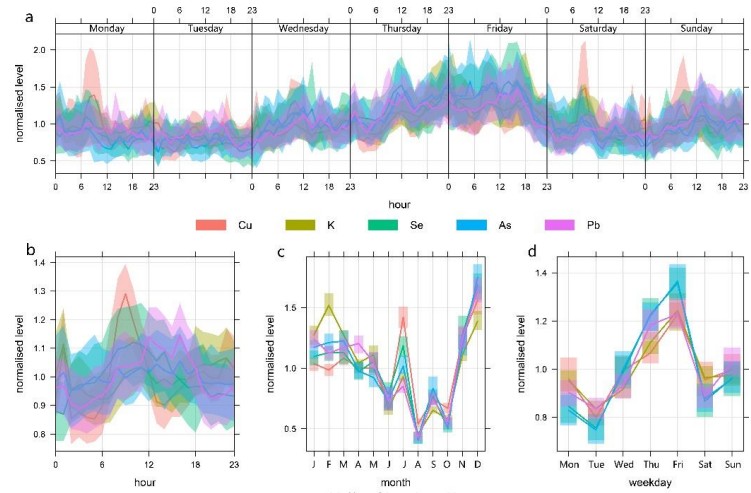


**Figure 10.** Weekly diurnal (**a**), diurnal (**b**), monthly (**c**), and weekly (**d**) variations of normalized Cu, K, Se, As, and Pb concentrations in Shanghai.






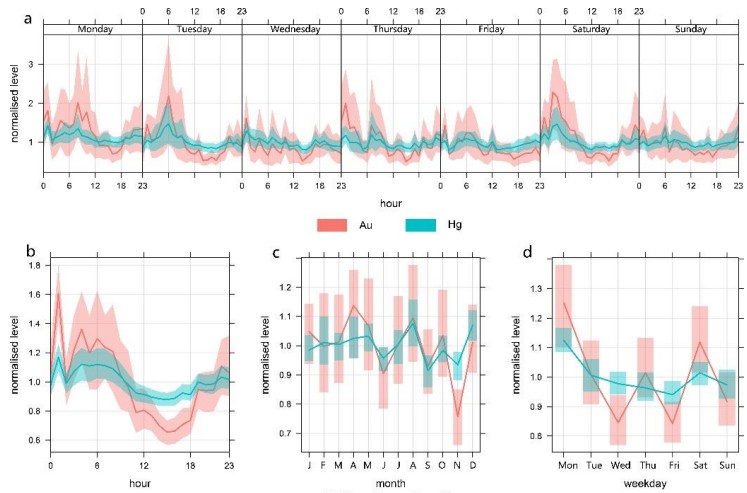

**Figure 11.** Weekly diurnal (**a**), diurnal (**b**), monthly (**c**), and weekly (**d**) variations of
normalized Au and Hg concentrations in Shanghai.







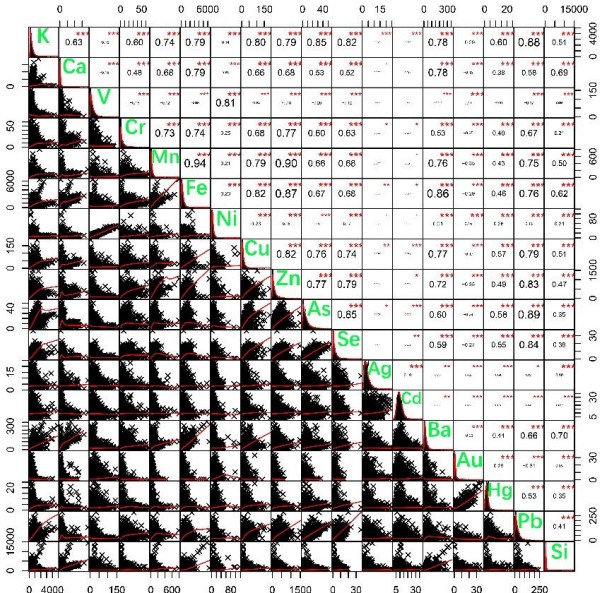

**Figure 12.** Spearman correlation matrix of 18 atmospheric elemental species in Shanghai between March 2016 and February 2017. The distribution of each species is shown on the diagonal. On the bottom of the diagonal, the bivariate scatter plots with a fitted line are displayed; on the top of the diagonal, the value of the correlation plus

the significance level as asterisks. Each significance level is associated to a symbol: *p*-values (0, 0.001, 0.01, 0.05, 0.1, 1) = symbols ("***", "**", "*", ".", " ").






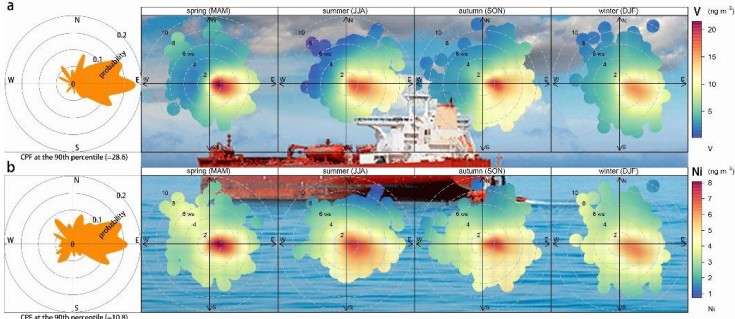

**Figure 13.** Conditional probability function analysis (left) and bivariate polar plots (right) of seasonal concentrations (ng m$^{-3}$) of V (**a**) and Ni (**b**) in Shanghai between March 2016 and February 2017. The center of each plot (centered at the sampling site) represents a wind speed of zero, which increases radially outward. The concentrations of V and Ni are shown by the color scale.



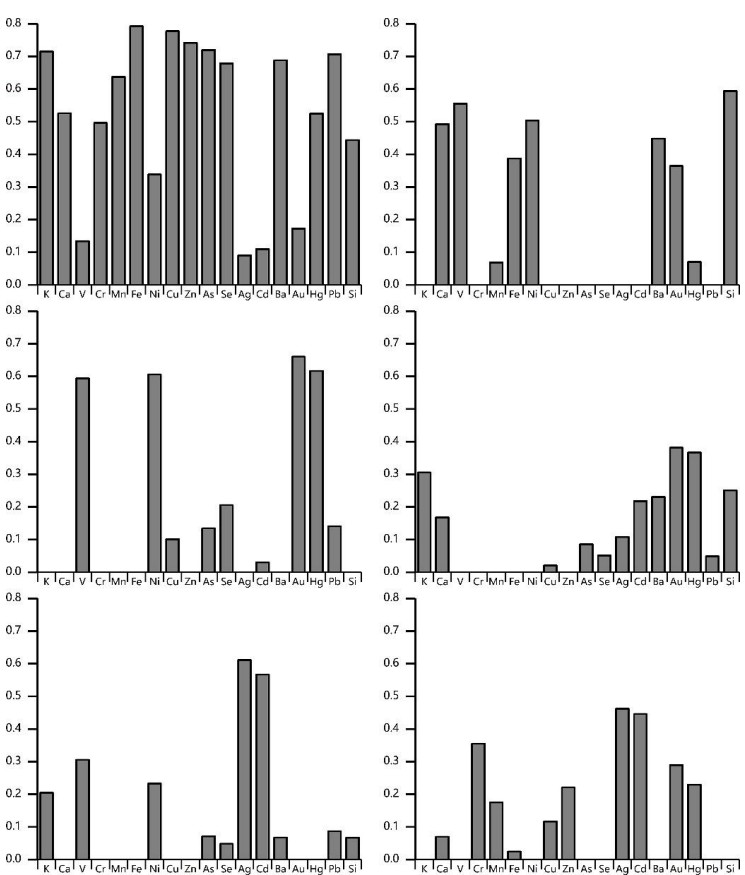


**Figure 14.** Principle component analysis of the 18 trace metals concentrations measured in Shanghai between March 2016 and February 2017. Six dominant factors are identified.





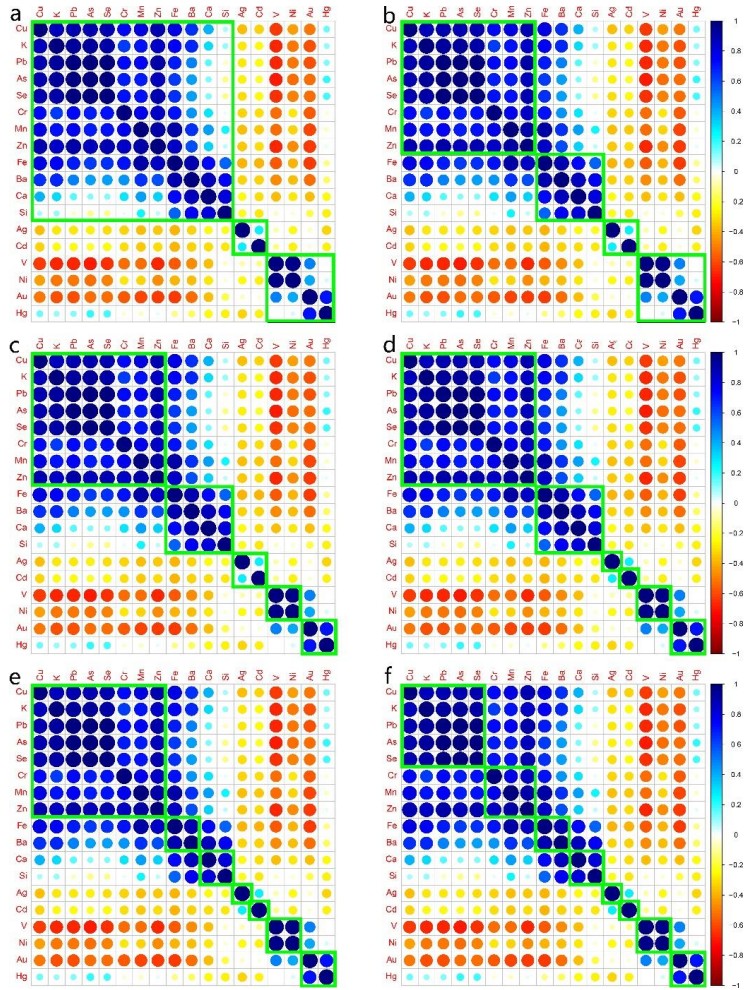

**Figure 15.** Hierarchical clustering orders of correlation matrix for the 18 trace elements measured in Shanghai between March 2016 and February 2017. Six solutions, from 3 factors to 8 factors, are shown from **a** to **f**.





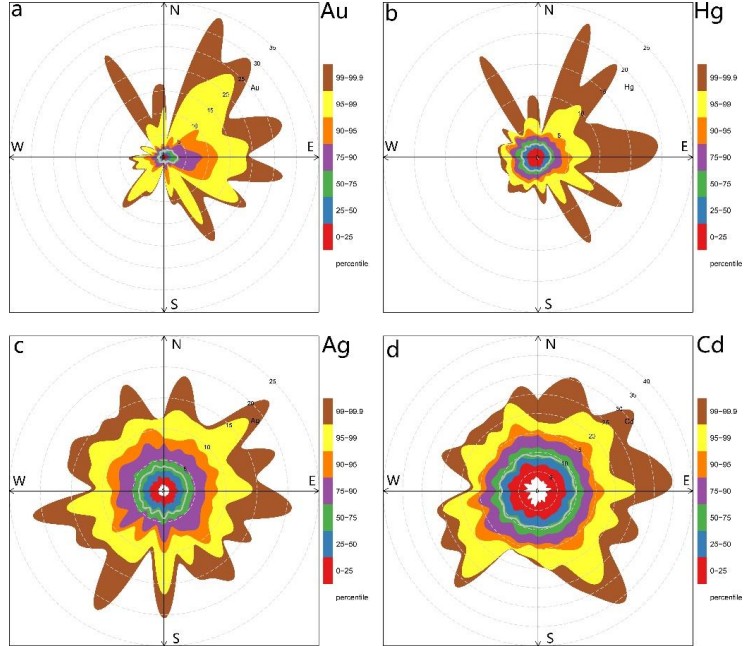

**Figure 16.** Percentile rose plot of Au (**a**), Hg (**b**), Ag (**c**), and Cd (**d**) concentrations in Shanghai between March 2016 and February 2017. The percentile intervals are shaded and shown by wind direction.








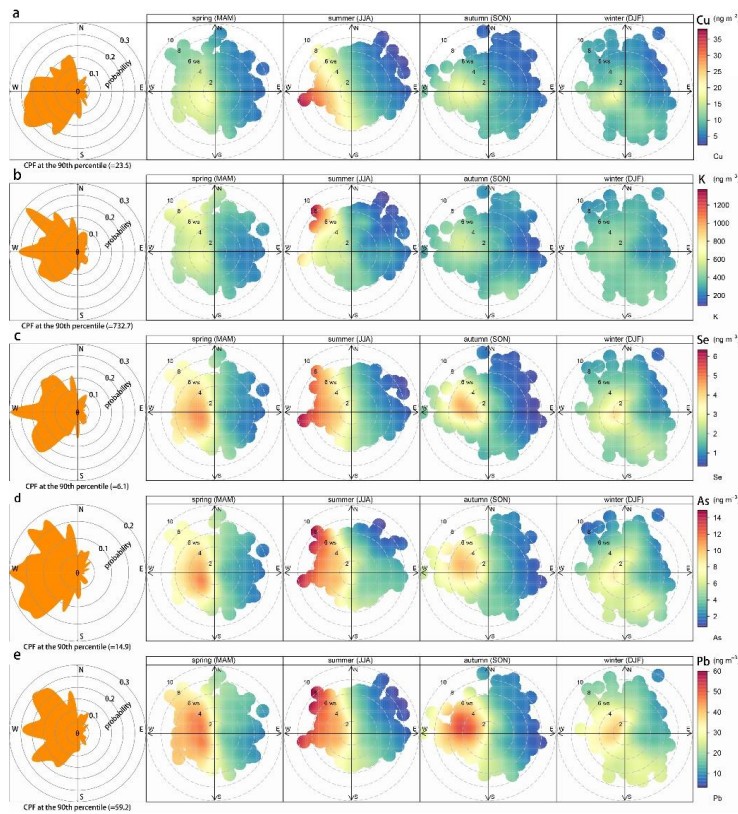

**Figure 17.** Conditional probability function analysis (left) and bivariate polar plots (right) of seasonal concentrations (ng m$^{-3}$) of Cu (**a**), K (**b**), Se (**c**), As (**d**), and Pb (**e**) in Shanghai between March 2016 and February 2017. The center of each plot (centered at the sampling site) represents a wind speed of zero, which increases radially outward. The concentrations of each species are shown by the color scale.
