# Peer review of "First long-term and near real-time measurement of trace elements in China's urban atmosphere: temporal variability, source apportionment, and precipitation effect"

_Atmospheric Chemistry and Physics, 2017_

## Referee Comment (RC1) · Anonymous Referee #1 · 3 Aug 2017

General comments

The manuscript discusses 1-year continuous measurements of 18 major and trace elements with an online XRF spectrometer in Shanghai, China. The authors argue that some trace elements are affecting human health in various ways, and that knowing the sources and behavior of trace elements will help in reducing these risks. The high time resolution of the measurements (1 hour) enables detection of concentration spikes, with their short-term acute exposure of humans. The dataset is analyzed with various statistical methods to attribute possible sources to the different elements and combinations. This is the first published year-long measurement of PM2.5 metals with

hourly time resolution.

The structure of the manuscript, the results and the presentation of the material are good. The topic is relevant and well worth publication in ACP. There are, however, a few changes and additions required before publication.

Specific comments

Considering the traditional methods of highly time-resolved trace element sampling and analysis it is stated (L121), '. . .they require a large commitment of analytical time.' This is not fully precise. Analyzing a sample may require only a few seconds (20 s up to a few minutes), which is quite fast. The problem is rather to get access to the accelerator facilities and enough beamtime to analyze thousands of samples of one single field campaign. Compared to wet-chemical techniques like ICP-MS, PIXE/SR-XRF is not so bad with respect to analysis time. Maybe some rewording might clarify this point.

In Section 2.1.1 the climatological description should be extended to better understand the seasonal variations of the data. Only the winter is characterized so far. To explain seasonal concentration statistics, the meteorological data should also be split seasonally, especially the wind data and precipitation. How is precipitation distributed during the full year of measurements? Does precipitation produce substantial cleansing of the polluted atmosphere? Is there a seasonal wind pattern (e.g. monsoon flow), or is the wind more or less equally distributed over the year? This might be relevant for explaining the origin of the coal combustion emissions in Fig. 17. The discussion of element concentration variations in Section 3.1.2 will also benefit from more climate information.

Another point is the statistics of the wind data, as shown in Figures 6 and S1. It is unclear how the wind data were processed for statistical analyses. As wind is a vector quantity, averaging and statistics should be done component-wise for u (east-west) and v (north-south) components. Averaging of wind directions may lead to stupid results, e.g. 350 and 10 degs would arithmetically average to 180 degs (a south wind), meteorologically however to 0 degs (a north wind). It is not evident from the description, whether the statistics packages used handle wind data correctly. From Figs. 6d and S1 the small variation of wind direction over 1 year appears doubtful, unless strong channeling of the wind had occurred. Furthermore, it is not clear how the wind direction as a circular distribution has been normalized. Here, a short explanation of the normalization procedure used would be helpful, also for interpreting the figures 5, 6, 8, 9, 10, 11.

The comparison of the Xact with filter data described in Section 2.1.2 appears not fully plausible. Of the 48 filters, 8 pairs of glass and cellulose filters were sampled concurrently. They are compared in Table S1, which indicates that the glass filters undersample by 10 to 25 % the aerosol relative to the cellulose filters (column 4). In column 5, the (remaining?) 40 filters were analysed together, relative to the Xact. The variation of the slopes is much larger in this case. Here it would be helpful to distinguish between the glass filters and the cellulose filters to see the effect of the filter type on the regression with the Xact. How are the regressions between Xact and filters for the 8 filters (two groups) individually that were pairwise sampled? I suggest to do the following analyses: Xact vs. glass fiber filters for the 8 filters, Xact vs. cellulose filters for the 8 filters, Xact vs. glass fiber filters for all glass fiber filters, Xact vs. cellulose filters for all cellulose filters, and then perhaps Xact vs. all filters. The argument for the higher slope of the Cr regression is not plausible, as Cr shows a significantly lower background value on cellulose than on glass (ratio 0.19), while Ba shows a ratio of 0.65, which is close to the average ratio cellulose/glass of 0.59. Cd, the third steepest slope, does also not indicate exceptionally high background values. If the background values were relevant, I would expect a shift of the regression line (i.e. a larger intercept), but not a change in slope. Therefore it is important to know the slopes for both cellulose and glass filters separately to infer something about the influence of background values. I propose to add the respective columns to Table S1, and to consequently distinguish the two filter types in the regression analyses.

The discussion of elements in Fig. 4 might be improved by a quantitative definition of the term 'seasonal variation'. In absolute concentrations, Ca and Si may show the highest degree of variation, but when normalized to their means and standard deviation, other elements might show a stronger relative variation. Si does also not show any variation between spring and summer. Which seasons are compared/considered for the seasonal variation range? Concerning the diurnal variations in Figs. 8 and 9, how would you explain the minimum occurring on Tuesday, i.e. 2 days after the weekend?

Another aspect is long-range transport, as discussed for V and Ni. The text states that based on Fig. 6 V and Ni are the result of mid- to long-range transport. However, Fig. 13 shows that V and Ni are rather local to mid-range transport from the southeast sector, which indicates that the relevant ship emissions originate both locally from the harbour on the Huangpu River and farther away from the sea east of Shanghai. It would be helpful to clarify what mid- and long-range transport means, i.e. which distances are attributed to this terminology. Does the ship traffic on the Yangtze River not contribute to the V and Ni concentrations observed at PEMC? Why is there not a larger contribution from the northeast? From Fig. 6 it is not clear how the wind direction varies in the course of the day. It appears as if the diurnal development of the atmospheric boundary layer overrides the development of a sea breeze system, if one exists in Shanghai. Fig. 6c also indicates an annual variation of wind direction that would override diurnal wind variations.

The issue with coal combustion is exciting. The authors show that coal-fired power plants are distributed evenly around the receptor site, while non-ferrous metals production is mainly in the west. These observations are illustrated with Hg, Au, and later with Cu, K, Pb, As, Se. What is the reason for such differences in metals emissions between power plants and metals production plants? Are there different cleansing systems in effect?

Technical corrections

[Figure]

The Lxxx indicate the line number xxx of the manuscript where a correction should be made.

L19 delete 'with', replace 'its' with 'their' L35 'orders of magnitude' (insert 'of') L39 delete 'were' L42 . . . was due to the interplay. . . L51 . . . combustion of coal. . . L64 delete 'John'. The correct reference is Duffus, 2002. L125 write 'distance-based detection in a multi-layered device'. This probably captures the essence of the technique better. L133 Suggest to add the references to Park et al. (2014) and Furger et al. (2017, Atmos. Measurement Techn., doi:10.5194/amt-10-2061-2017) here, as both papers discuss the data quality of the Xact-625. L134 Add (YRD) as an abbreviation for the Yangtze River Delta. L173 . . . from Siberia which can cause. . . L186 . . . such as V. . . L258 . . . to do correlation matrix calculations. I am unsure what is missing here. L289 The two methods have been implemented in the . . . L295 Replace 'and give the probability of doing so' with 'and with which probability.' L327ff The different limiting values would probably be easier digested when listed in a Table. L338 replace 'with' by 'while' L375 please give a reference. L385 replace 'more' by 'other', and 'shown' by 'showed'. L404 . . .in Shanghai has occurred during Sunday (Fig. 6d) (February (Fig. 6c)). – refer to the correct sub-figures. L415 'pads' instead of 'pad'. L416 correct 'less traffic flow in weekends not only lower road suspend dust but also cut metal species emissions' to 'less traffic flow in weekends not only lowers re-suspended road dust but also reduces metal species emissions'. L418 replace 'Ca' by 'Si'. L419 replace 'July' by 'June' and Fig. 10 by Fig. 8. Then replace 'Si' by 'Ca'. Please correct also the remark on the 0100 h Si peak. Do you have an explanation for this huge peak after midnight? L433 'transforming' – do you mean 'transporting'? L488 The text mentions 4 statistically significant factors. Which of the 6 factors in the figure are these? You should indicate the significance for all 6 factors in the Figure. L498 Replace 'were' by 'are'. L514 'plot' instead of 'plots'. Delete 'is'. L545 delete 'that'. L549 delete 'is' L553 write 'combustion of coal is located;' L580 write 'Fig. S4 also evidently reflects that high concentrations of Zn can occur in the northwest of PEMC.' L698 write: 'Duffus, J. H.: . . ..'

L1066 A land use map (not land uses map) L1115 The grey line indicates one two standard deviations – one or two? Fig. 6 Indicate the wind direction axis (or explain normalization of wind direction, see remark above). L1224 Replace 'On the bottom' with 'Below the diagonal'. L1225 Replace 'on the top' with 'above'. L1247 Write 'Principal' instead of 'Principle' Fig.15 Explain the circle sizes. Table S1: Caption: different monitoring (x, y) should be indicating the correct, correlated quantities, probably y Xact, x filter.

---

## Referee Comment (RC2) · Anonymous Referee #2 · 27 Aug 2017

Using an on-line x-ray fluorescence system (XACT), this paper describes measurements of trace elements in PM2.5 aerosol in Shanghai, the industrial center of China. This pioneer work is informative and valuable in terms of the number of species (18) and the duration of observations (a year cycle with hourly resolution) presented, which permit a thorough analysis of temporal variations and source attribution. Meanwhile, the authors do an extensive validation of the performance of XACT from solution of filter-based metals measurement which they extract a correction factor applied to their data. Overall, this is a nice work and I recommend it for publication.

Comments: The technical aspects were adequately covered by the referee 1. Particularly, I am also interested in seeing if precipitation produces substantial scavenging of airborne trace metals. Several typical events with strong precipitation can be selected as a better way, in my opinion, to examine the effect of wet removal in the supplement.

Title: Given that the focus of the MS is trace metals, I suggest the title can be changed as "First long-term and near real-time measurements of atmospheric trace metals in Shanghai, China".

Table 1: Data of Shanghai should be listed adjacent to that of Gwangju.

Fig. 1: I didn't saw (a), (b), (c), (d), and (e) in the figure. Moreover, please specify the data sources of population density and major point sources.

Fig. 6: It could be a problem that the variations of wind directions and the mass concentrations of V and Ni are normalized together. I suggest the authors to re-plot Fig. 6 by excluding wind directions.

Line 236: Is there necessary to use data like CO, NO2, and SO2 in the study?

Line 457: 3.2.1 Pinpoint the most possible source.

---

## Referee Comment (RC3) · A.J. Dore (Referee) · 29 Aug 2017

Review of: "First long-term and near real-time measurement of atmospheric trace elements in Shanghai, China" by Yunhua Chang et al

Overview The rapid development and industrialisation in China during recent decades has resulted in increased emissions of many pollutants leading to high levels of particulate matter in the atmosphere with severe effects on human health. Amongst the pollutants emitted, heavy metals are of particular concern due to their toxicity at high air concentrations. The problem is of specific regional concern as in other parts of the world (i.e. many European countries) concentrations of heavy metals in the at-

mosphere have decreased dramatically since their peak values to values which are now well below limit values considered to pose a threat to human health. The study focuses on Shanghai which is a mega-city and a centre for heavy industry. The paper suggests that metal concentrations are one or two orders of magnitude higher than in north America and Europe which is a highly significant conclusion as this indicates that a large population is exposed to multiple and serious threats to human health in the region. An extensive data set has been analysed which includes both a large number of species and high frequency measurement. The chemical analysis covers 18 heavy metal species which allows grouping of metals into different source categories. The hourly measurement of the data is of particular significance as such high temporal resolution of measurements for a full year of monitoring combined with detailed chemical analysis is quite rare and allows analysis using the conditional probability function and bivariate polar plots. The manuscript is well referenced and the text is logically constructed. The experimental set up is clearly described and graphical representation is put to good use to investigate seasonal, weekly and diurnal variation in metal concentrations. Some of the plots (i.e. figures 2 and 12) have details which are in very small print. However I think this is probably necessary due to the large number of metals to be simultaneously displayed and, as the graphs are of high resolution, the information can be read easily using the zoom function. I have a few queries about interpretation of the results which are raised below. The English is of a reasonable standard. However the language needs checking as there are various minor grammatical errors in the text (i.e. use of the article a / the) and some inappropriate / unclear expressions for a scientific publication, a few of which (in the early part of the text) are mentioned below. I am pleased to recommend the paper for publication following attention to the comments below.

Examples of minor corrections: Abstract: - " . . .with poorly constrained on its abundances and sources . . ." Change to: " . . . which have considerable uncertainty associated with the source and magnitude of their atmospheric emissions"

- "collocated" should be "co-located"

- "Besides, various mathematical methods and physical evidences were served as criteria to constrain various solutions of source identification." Change to: "A variety of mathematical techniques were employed with high frequency monitoring data to identify sources for metal emissions."

Main text

- "Among the chemical components relevant . . ." Long sentence. Break into two. - Line 70 – 80. Health effects of some metals are mentioned here as human carcinogens. However there are many other individual effects on human health (i.e. brain, lungs, reproduction, kidneys(?) for individual metals which could also be mentioned here. Recent American studies have even linked Pb concentrations to criminal behaviour.

- Line 135: ". . . Shanghai is home to nearly 25 million people as of 2015, marked as the largest megacity in China" should be ". . . making it the largest mega-city . . ."

- Line 150: "Meanwhile, most available source evidences were inferred based on filter sampling and off-line analysis, which were not necessarily representative of actual origins." The statement is unclear and needs re-wording

- Line 184: change 'multitudinous' to 'a multitude of'

- Line 199 'resulted' should be 'resulting'

- Line 203 'producing' should be 'produced'

- Line 246: 'were followed' should be 'followed'

- Figure 3 cation: 'A quick glance of the mass concentrations of 18 trace elements measured . . .'. Change to 'Mass concentrations of the 18 trace elements measured . . .'

- Line 338 'which accounting for' should be 'which accounts for'

- Line 362: 'because certain emission sources may make a pause or reduction during weekends' Change to 'because certain emission sources may be paused or reduced during weekends'

- Figure 4: 'The gray line indicates one two standard deviations.' Needs correcting

Specific comments

- To me the title, whilst accurate, is a little awkward. I think 'first' is unnecessary and could be removed as we can argue that all scientific publications are in some way a first. How about "High frequency monitoring of atmospheric trace elements in Shanghai, China"? 'Monitoring' implies long term measurement. A longer version could be: "High frequency monitoring of atmospheric trace elements in Shanghai, China, and source attribution analysis"

- Mean and variance values of concentrations are quoted to 4 or even 5 significant figures. I think it is unlikely that this degree of accuracy can be attributed to the measuring system. 2 significant figures (or a maximum of 3) would be more realistic.

- Figure 1: Include (a) , (b), (c), (d), (e) next to individual figures

- Line 402: "This can be used to explain that the weekly (monthly) lowest impact of shipping emissions in Shanghai was occurred during Sunday (Fig. 6c) (February (Fig. 6d))." Check this statement. February doesn't appear from Figure 6(b) to be the month with the minimum concentrations

- Line 410: "As demonstrated in Fig. 8d and Fig. 9d, there is an evident drop in the concentrations of Si, Ca, Fe, Ba, Mn and Zn after entering weekends." The minimum concentration in Figures 8(d) and 9(d) appears to be on Tuesday. Can the two day delay between the weekend and peak concentrations be explained? If this is following a weekend dip in activity then perhaps it suggests long range transport?

- Wind speed is associated with long range transport in the text. However high wind speeds also cause production of sea salt aerosol and re-suspension of surface dust

particles, both of which can contain heavy metals and base cations. The text refers to high concentrations of metals in top soil but doesn't include correlation of metals concentrations in air with high wind speeds. In some European countries, re-suspension of surface dust makes a major contribution to heavy metal concentrations in air which can be even higher than primary emissions (see i.e. Travnikov et al, 2012 and other reports from MSC-east). I suggest that re-suspension of surface dust during high winds is considered in the analysis of the relationship between metal concentrations in air and wind speed / wind direction.

Travnikov O, Ilyin I, Rozovskaya O, Varygina M, Aas W, Uggerud HT, et al. Long-term changes of heavy metal transboundary pollution of the environment (1990–2010) EMEP contribution to the revision of the Heavy Metal Protocol; 2012 [Available at: http://www.emep.int/publ/
* * *

---

## Referee Comment (RC4) · Anonymous Referee #5 · 3 Sep 2017

This paper presents a full year of temporally highly resolved data (1h resolution) of trace elements in PM2.5 as measured in Shanghai (China) using an on-line multi-metal monitor (XRF instrument Xact625, Cooper Environmental). This data is evaluated by applying different statistical methods in order to find correlations between the different elements and evidence of common sources. The article presents probably the longest time series of hourly trace elements in PM2.5 so far, and highlights the situation in the largest megacity and a main industrial center in China, where emissions are expected to be high. The article is therefore of high interest, it is reasonably well written (see comments below) and certainly deserves publication in Atmospheric Chemistry and Physics. However, the manuscript needs some re-visions as detailed in the comments

below:

1. Abstract: The abstract is much too long and needs to be shortened. Only the most important results should be mentioned in a short and concise way.

2. Although not a native English speaking person, I find the English is of variable quality. Abstract and introduction are well written, the other sections need careful revisions. Especially the Statistical analysis section has many linguistic and semantic errors. Examples are the lines 256-258, "The "corrplot" package in R is a graphical display of a correlation matrix, confidence interval. It also contains specific algorithms to do matrix.", and lines 278-279, "The corrplot package can draw rectangles . . .". Please revise the text carefully.

3. The results of the principal component analysis and also the hierarchical clustering do not bring any new insights that go beyond the analysis of pairwise correlations together with CPF and BPP. Section 3.2.2 is not well written and does not give any new information, the discussion about the number of selected clusters appears arbitrary. Moreover, the discussion of the method applied for hierarchical clustering is not sufficient. It remains unclear what exact method has been applied. If it is decided that the results of this data analytical method can remain in the article, then a detailed description of the applied method is required. I suggest to skip section section 3.2.2, as it does not provide any new insights and it is also not well written. The paper will certainly benefit from being shortened (the authors could mention in the paper that PCA and hierarchical clustering were applied, but the results did not lead to additional information).

4. Figure 5 is rather suspicious. The sharp peaks in Cd and Ag at 1am local time are hard to believe. The authors should check again, if there is not another, maybe operational explanation. In Figure 6, normalised wind does not make much sense, should be removed from the graph. Figures 8-11 are of the same type than Figures 5 and 6. They can be shifted to the Supplement.

Additional comments:

Lines 98-99, Sentence should be changed to e.g. "Typical ambient trace metal sampling devices collect 12 to 24-hr integrated average samples, which . . ."

Line 105, "may be orders of magnitude lower than . . .". It can easily been estimated that it cannot be orders of magnitude or max 2 orders for 15min samples. Rewrite accordingly.

Lines 152-153, "Meanwhile, most available source evidences were inferred based on filter sampling and off-line analysis, which were not necessarily representative of actual origins." I don't understand this sentence, it is probably not correct. Please rewrite.

Line 219, please be more precise, should be "glass fibre filters".

Line 229, should be "spectrometer".

Lines 242-244: "As data in the current study were collected in near-real time, the importance for quality assurance and quality control (QA/QC) system can be crucial in order to improve data quality throughput." All measurements independent of time resolution require an appropriate QA/QC. This sentence makes in the present form no sense and should be deleted or revised.

Lines 294-296, sector. "CPF analysis is capable to 295 show which wind directions are dominated by high concentrations and give the probability of doing so." Poor English, please rewrite.

Lines 330-331: The statement that "airborne metals pollution in Shanghai is generally low by the current limit ceilings" is difficult to see from the text, maybe add a table (could also be in the Supplement).

Lines 373-375. "Globally, anthropogenic emissions of Ag and Cd exceed the natural rates by well over an order of magnitude*. Please give a reference for this statement.

Line 562, legend of Figure 12, and at elsewhere in the text. The authors mention "significant corre-lations". It should be explained how significant correlations are defined, how has significance been calculated, what kind of statistical test has been applied.

---

## Author Comment (AC1) · 30 Nov 2017

Thank you for carefully reading the manuscript and providing useful suggestions to improve the paper. The letter of reply and the changes can be found in the pdf document.

Please also note the supplement to this comment: https://www.atmos-chem-phys-discuss.net/acp-2017-613/acp-2017-613-AC1-supplement.pdf

2017.

---

## Author Response (AR1)

We greatly thank the reviewers for their insightful comments, which we've addressed (in bold) point by point in our reply letter below. The revised MS is attached after the reply letter. We are sorry for the delay due to the family affairs of the first author.

**Anonymous Referee #1**

**General comments**

The manuscript discusses 1-year continuous measurements of 18 major and trace elements with an online XRF spectrometer in Shanghai, China. The authors argue that some trace elements are affecting human health in various ways, and that knowing the sources and behavior of trace elements will help in reducing these risks. The high time resolution of the measurements (1 hour) enables detection of concentration spikes, with their short-term acute exposure of humans. The dataset is analyzed with various statistical methods to attribute possible sources to the different elements and combinations. This is the first published year-long measurement of PM2.5 metals with hourly time resolution. The structure of the manuscript, the results and the presentation of the material are good. The topic is relevant and well worth publication in ACP. There are, however, a few changes and additions required before publication.

**Many thanks for the favorable comments and constructive suggestions.**

**Specific comments**

Considering the traditional methods of highly time-resolved trace element sampling and analysis it is stated (L121), '... they require a large commitment of analytical time.' This is not fully precise. Analyzing a sample may require only a few seconds (20 s up to a few minutes), which is quite fast. The problem is rather to get access to the accelerator facilities and enough beam time to analyze thousands of samples of one single field campaign. Compared to wet-chemical techniques like ICP-MS, PIXE/SR-XRF is not so bad with respect to analysis time. Maybe some rewording might clarify this point.

Fully agree. The sentence '... they require a large commitment of analytical time' has been deleted in the revised MS.

In Section 2.1.1 the climatological description should be extended to better understand the seasonal variations of the data. Only the winter is characterized so far. To explain seasonal concentration statistics, the meteorological data should also be split seasonally, especially the wind data and precipitation. How is precipitation distributed during the full year of measurements? Does precipitation produce substantial cleansing of the polluted atmosphere? Is there a seasonal wind pattern (e.g. monsoon flow), or is the wind more or less equally distributed over the year? This might be relevant for explaining the origin of the coal combustion emissions in Fig. 17. The discussion of element concentration variations in Section 3.1.2 will also benefit from more climate information.

We appreciate the useful comments, in which the potential effect of wet cleaning on the mass concentrations of trace elements had also been mentioned by referee #2.

1) How is precipitation distributed during the full year of measurements?

The figure below shows the variations of precipitation at hourly, monthly, and seasonal scales during the full year of measurements. It is clear that the amount of precipitation in winter is much lower than that of other seasons.

2) Does precipitation produce substantial cleansing of the polluted atmosphere?

The short answer is that precipitation produced different cleansing effects on the measured trace elements. In the whole 8760 hours (W in short), there were 1078 hourly rainfall events (R in short) occurred during the full year of our measurements. Here the average mass concentrations of a given species for W and R are defined as  $W_a$  and  $R_a$ , respectively. The ratio of  $W_a$  higher than  $R_a$  ([ $W_{a}$ - $R_a$ ]/ $R_a$ \*100) for the 15 trace elements is illustrated in the figure below, in which 6 of them (denoted in red) are close or lower than 0, suggesting that they were not influenced by wet scavenging. Note that the  $R_a$  for trace elements of K, Cr, and As are not included due to the number of the data points is lower than 719 (two thirds of the 1078 hourly rainfall events).

3) Is there a seasonal wind pattern (e.g. monsoon flow), or is the wind more or less equally distributed over the year? This might be relevant for explaining the origin of the coal combustion emissions in Fig. 17. The discussion of element concentration variations in Section 3.1.2 will also benefit from more climate information.

As we mentioned in the MS that Shanghai has a humid subtropical climate and experiences four distinct seasons. Winters are chilly and damp, with northwesterly winds from Siberia can cause nighttime temperatures to drop below freezing. In summer, airflow carries moist air from the Pacific Ocean to mainland China. The city is also susceptible to typhoons in summer and the beginning of autumn.

We agree that our discussion related to the explanation of potential sources should be divided into two dimensions: emission sources and winds direction. Specifically, in Fig. 17, the potential source regions of Cu, K, Pb, As, and Se are identified (shown as below), in which all species are generally transported from far western Shanghai. From the perspective of emission sources, the activities of stationary combustion of coal were concentrated in western Shanghai. Besides, although winds direction in Shanghai varies distinctly in four seasons, western Shanghai was constantly as the most possible source region. Collectively, we concluded that ambient concentrations Cu, K, Pb, As, and Se were mainly originated from coal combustion.

---

## Author Response (AR2)

Dear Prof. Willy Maenhaut,

We thank you for handling our paper with your valuable time and expertise. After careful consideration of your comments, a major revision has been made to thoroughly address all your concerns.

In the revised MS (after the response letter), words marked in red indicate the contents in the original MS that have been deleted, and words marked in green (mainly including the results of PMF analysis and precipitation scavenging effect) indicate the new contents that we've added.

Below please see our point-by-point reply to your comments. We have tried to answer as detailed as possible.

**Comment 1:** The authors should have addressed the comments of the referees in a better way. Their reply to the second specific comment of Anonymous Referee #1 (with regard to the precipitation) makes in my opinion no sense. The approach that was used to handle this comment is much too simplistic. It is impossible that the atmospheric concentrations of elements such as V, Ni and Si are not influenced by wet scavenging.

**Reply:** We've made a great effort to examine the potential scavenging effect induced by precipitation on the mass concentration of ambient trace elements from the perspective of their species and sources. In brief, we found that (1) although precipitation is effective in term of reducing the overall levels of ambient trace elements, there is no uniform pattern for each elemental species; (2) there is a threshold of precipitation amount to lower ambient $PM_{2.5}$ trace elements mass; (3) trace elements contributed by sources of traffic-related (exhaust and non-exhaust) and coal combustion can be significantly removed from the atmosphere during precipitation, while this is not the case for the sources of ferrous (iron and steel) and nonferrous (Cu, Pb, Zn) metal smelting; (4) interestingly, the variations of elements concentrations contributed by shipping (mainly V and Ni) before, during, and after precipitation events were mainly

governed by wind speed and direction instead of precipitation. For more details, please refer to section 3.3 in our revised MS:

[revised manuscript text omitted]

**Comment 2:** Overall, the current version of this manuscript still has serious shortcomings. 1. Although the language and grammar of the manuscript are reasonable, there is still a lot of room for improvement. Several alterations are suggested under "Further comments" below.

**Reply:** Thank you for your careful revision. All mistakes you mentioned regarding language and grammar in the original MS have been revised accordingly.

**Comment 3:** 2. The authors followed up on the suggestion of Anonymous Referee #2 to replace the word "elements" in the title by "metals". I disapprove of this change. Three of the 18 elements measured (i.e., Si, As and Se) are not metals. Si and As are metalloids and the nonmetal Se is also sometimes considered a metalloid. Also throughout the text "metal(s)" is often used where it should be "element(s)".

**Reply:** Agree. The full text of the MS has been thoroughly checked.

**Comment 4:** 3. I agree with Anonymous Referee #5 where he states that the results of the principal component analysis and also the hierarchical clustering do not bring many new insights. Still there are many sentences in the revised manuscript devoted to the results of those two approaches. It is unclear to me why did the authors did not make use of Positive Matrix Factorization (PMF) for source identification and apportionment.
thoroughly

**Reply:** Agree. The results of the principal component analysis and the hierarchical clustering have been deleted in the revised MS. Alternatively, 
[revised manuscript text omitted]

**Comment 5:** 4. The concentration data in lines 28-31 and in lines 316-319 (and on some other occasions in the manuscript) contain too many significant figures. Two significant figures suffice in case the first significant figure is larger or equal than 2, and when the first significant figure equals 1, three significant figures can be used.

**Reply:** Revised accordingly.

**Comment 6:** 5. Section 2.2.1: I presume that hourly concentration data were used in the statistical analyses. If so, this should clearly be indicated.

**Reply:** Indicated as your suggestion in the revised MS.

**Comment 7:** 6. Lines 479-480: Considering that up to nearly 9000 data points were available for each element (if hourly concentration data were used in the statistical analyses), how can the authors make the statement that "much larger datasets are required for definitive source apportionment with PCA"? Even in case daily averages were used in the PCA, then there would still be sufficient data for an appropriate PCA.

**Reply:** Agree. PCA analysis has been deleted in the revised MS.

**Comment 8:** Further comments for the Main text:

Lines 26 and 126: What is meant by "reactive filter"? In line 193 it is mentioned that a Teflon filter tape was used in the Xact instrument. Teflon filters are typically not reactive at all.

**Reply:** Sorry for the confusion. There "reactive filter" essentially means "moving filter" because filter can be automatically advanced into the analysis area. To avoid misunderstanding, "reactive filter" has been replaced as "Teflon filter".

**Comment 9:** Line 34: Replace "with other" by "with that in other".

Line 35: Replace "in sites" by "at sites".

Line 44: Replace "sourced from" by "derived from".

Line 213: Replace "at PMC" by "at the PMC".

Line 226: Replace "for species of" by "for".

Line 228: Replace "than other" by "than those of other".

Line 248: Replace "principle component" by "principal component".

Line 257: The acronym "PCA" was already defined in line 248. Its should not be defined again.

Line 303: Replace "has described" by "has been described".

Line 305: Replace "3 Results" by "3. Results".

Line 310: Replace "illustrated in" by "presented in".

Line 334: Replace "accounting for" by "accounts for".

Line 356: Replace "willalso" by "will also".

Line 363: Replace "were also illustrated in" by "are shown in".

Line 378: Replace "winter were" by "winter is".

Line 383: Replace "later)" by "below)".

Line 387: Why is reference made here to the wind speed? There is no information about the wind speed in Fig. 6c.

Line 394: Replace "can be served" by "can serve".

Line 400: Replace "well as" by "well as for".

Line 402: Replace "evening" by "in the evening".

Line 414: Replace "from other" by "from that of other".

Line 415: Replace "and 16:00" by "and at 16:00".

Line 416: Replace "were primarily" by "are primarily".

Line 417: Replace "were more" by "are more".

Line 421: Replace "matters" by "matter".

Line 439: Replace "be primarily sourced" by "primarily originate".

Line 441: Replace "of Cu origins" by "of the Cu origin".

Line 454: Replace "and presented" by "and they are presented".

Lines 456-457: Replace "trace crustal matters" by "crustal matter".

Line 458: Replace "trace crustal matters" by "the crustal matter trace elements".

Line 461: Replace "trace crustal matters" by "crustal matter elements".

Lines 471-473: There is something wrong with the wind directions given here. It seems to me that "west" in line 471 should be replaced by "east" and "southwest" in line 473 by "southeast".

Line 473: Replace "are sourced from " by "originated from".

Line 478: Replace "Explore and constrain more" by "Exploration and constraining of more".

Line 489: Replace "later)" by "below)".

Line 490: Replace "Matrix reorder" by "Matrix reordering".

Line 505: Replace "illustrated in" by "presented in".

Line 511: Replace "Even though" by "Nevertheless".

Line 517: Replace "illustrated in" by "shown in".

Line 529: It is unclear to me what is meant by "reaming".

Line 552: Replace "matters from" by "matter from".

Line 554: Replace "be sourced from" by "originate from".

Line 560: Replace "principle source" by "principal source".

Lines 573-574: Replace "approximately 20 km apart" by "at approximately 20 km".

Line 575: Replace "also evidently reflects" by "also indicates".

Line 597: Replace "e.g." by "i.e.".

Line 624: Insert ", Edinburgh, United Kingdom," after "Hydrology".

Lines 648-658: "Carslaw and Ropkins, 2012" should come before "Carslaw et al., 2006".

Line 844: Replace "Inventory" by "inventory".

Line 1001: Replace "A collection of" by "Overview of".

Line 1008: Replace "although a huge" by "a huge" and replace "had been" by "has been".

Line 1009: Replace "in which no" by "but that no".

Line 1017: Replace "A land use" by "Land use".

Page 34, Figure 2: The unit for the abscissa data ("value") in the right part of the figure should be specified.

Line 1034: Replace "A quick glance of the mass" by "Average mass".

Page 37, Figure 5d: Replace "weakday" by "weekday" in the abscissa.

Page 38, caption of Figure 6, line 2: There is no information about the wind direction in the Figure.

Page 44, Figure 12: The unit for the data in the abscissas and ordinates should be specified. The data in several cells for V, Ni, Ag and Cd are not readable, although for some of them a significance level of "***" is specified. Then, in the last line of the figure caption, 6 p-values are given and only 5 symbols; besides, it is unclear what the last of these 5 symbols really is.

Page 46, Figure 14: It should be indicated that this figure shows component loadings. Furthermore, in the caption of the figure "Principle component" should be replaced by "Principal component".

Page 48, Figure 16: The unit for the concentration data in this figure should be specified.

Comment for the Supplement:

Caption of Table S1: Replace "Inter-comparison of trace metals" by "Inter-comparison between ICP-MS and Xact for trace elements".

**Reply:** We were deeply touched by your careful revision, thanks!

All the comments above have been revised accordingly in the revised MS.

[revised manuscript text omitted]

---

## Author Response (AR3)

Dear Prof. Willy Maenhaut,

We thank you again for handling our paper and for your valuable comments. Addressing the points, you and two reviewers raised, we have prepared a revised manuscript with track-changes model (page 24-66). Please see our point-by-point reply to the comments of you (page 1-15), reviewer 6 (page 16-21), and reviewer 2 (page 22-23) below.

Co-Editor's Comments:

*The manuscript has been substantially improved with the PMF results. The aspect of precipitation scavenging is also presented in a better way, but there are still problems with it as indicated in the report of Anonymous Referee #6. The comments and corrections of both Anonymous Referees need to be fully addressed. I also have several comments and corrections, which are listed below, and need to be taken into consideration. Consequently, substantial revision is still needed before this manuscript can be accepted for publication in ACP.*

Main comments for the Main text:

1. *The language and grammar of the manuscript are fair, but there is very much room for improvement. Many alterations are suggested under "Further comments" below.*

**Reply**: Thank you very much for your careful revision. We had asked a native English speaker for the language correction before the submission, sorry for so many language problems still existed. All the problems you raised have been corrected in the revised MS.

2. *Line 41: The ratio of 0.31 is too low for a V/Ni ratio in shipping emissions. Does the 0.31 perhaps indicate the Ni/V ratio? If so, replace it by the V/Ni ratio. See also the comment of Anonymous Referee #6 on this topic.*

**Reply**: Sorry for our carelessness. The ratio of 0.31 should be "Ni/V" ratio. Given that "V/Ni" ratio is a well-known term as pointed out by Anonymous Referee #6, we revised

the text and figure accordingly. The previous figure was plotted by Openair package in R, and the revised figure was plotted by OriginPro 2017.

3. *Lines 215-216: Since the Micromatter standards were used for calibration, it is unclear to me how one can then derive independent measured and standard masses from them.*

**Reply**: No need to derive independent measured and standard masses. The Quantitative Aerosol Generator (QAG) creates an aerosol of known concentration by nebulizing a solution. The resulting droplets are carried out of generation area to a drying chamber where they are dried to salt particles. Depending on the application for the salt particles the size can be varied to a number mean between 0.1 and 2 μm in diameter. The aerosol exiting the drying chamber contains a known concentration of analytes that is calculated from the QAG's input parameters. This reference aerosol produced by the QAG is process traceable to the National Institute for Science and Technology standards and can be used to challenge and evaluate the accuracy, precision, and linearity of measurement methods such as the XACT metal monitor during sampling.

4. *Line 248: Abbreviations and acronyms (here "USEPA" and "QA/QC") should be defined (written full-out) when first used. Since the acronyms are only used here, "USEPA" can simply be replaced by "United States Environmental Protection Agency" and "QA/QC" by "Quality Assurance / Quality Control".*

**Reply**: Correct. We revised the text accordingly.

5. *Line 249: It is unclear what is meant by "upscale values".*

**Reply**: The term "upscale values" is essentially means "quality-assured values". The second phase of the quality-assurance (QA) run of the XACT metal monitor is the upscale sample. Specifically, the linear actuator is used to introduce a known sample into the analysis area containing a large amount of Cr, Cd, and Pb. A 15 minute XRF analysis of this sample is performed, and the metals data is sent over to the sample

controller. As this XRF analysis is taking place, the sample controller is also performing

60   the QA flow check. The QA flow check uses a different flow path and flow meter to
measure and confirm that the sample flow is within the allowed limits.

6.   Line 269: The phrasing of "with all elements missed" is somewhat unclear. I suggest
to replace "with all elements missed" by "with missing data".

65   **Reply**: Agree. We revised the text accordingly.

7.   *Page 13, within Figure 2 (and on various other occasions in the manuscript, e.g.,*
*in Table 1, data for this study; in line 334; within Figure 4; in line 408; in line 426):*
*The concentration data contain too many significant figures. Two significant figures*

70   *suffice in case the first significant figure is larger or equal than 2, and when the*
*first significant figure equals 1, three significant figures can be used.*
**Reply**: Agree. We decided to delete Figure 3 in the revised MS because it is redundant.
We keep Table 1 because it is necessary to perform data comparison with other relevant
studies. For line 408, we need to specify the mass fraction of several trace elements

75   markers, which cannot be directly reflected in Figure 4. Although the specific figures
have been shown in Fig. 4, we treated line 426 as a mini conclusion, and we want to
convey this important message to readers. Thank you for your understanding.

8.   *Page 20, Figure 6: Please, replace this figure by one with V in the ordinate and Ni*

80   *in the abscissa, and provide thus regression lines with V in the y-axis. See also the*
*comment of Anonymous Referee #6 on this topic.*
**Reply**: Agree. We revised the figure accordingly.

9.   *Lines 492-494: Literature references are needed for the statements in these two*

85   *sentences.*
**Reply**: Agreed; two studies that are relevant in this context have been added in the
revised MS.

Morawska, L., and Zhang, J.: Combustion sources of particles. 1. Health relevance and source signatures, Chemos., 49, 1045-1058, doi: 10.1016/S0045-6535(02)00241-2, 2002.

90

Tian, H. Z., Zhu, C. Y., Gao, J. J., Cheng, K., Hao, J. M., Wang, K., Hua, S. B., Wang, Y., and Zhou, J. R.: Quantitative assessment of atmospheric emissions of toxic heavy metals from anthropogenic sources in China: Historical trend, spatial distribution, uncertainties, and control policies, Atmos. Chem. Phys., 15, 10127-

95

10147, doi: 10.5194/acp-15-10127-2015, 2015.

*10. There are several problems with the References.*

**Reply**: Thanks again for your very careful revision. We've corrected all the problems you mentioned.

100

*11. The following references to which is referred within the text are not in the Reference list:*

*Line 256: Paatero and Tapper, 1994.*

**Reply**:

105 Pentti, P., and Unto, T.: Positive matrix factorization: A non-negative factor model with optimal utilization of error estimates of data values, Environmetrics, 5, 111-126, doi:10.1002/env.3170050203, 1994.

*Line 271: Kim et al., 2005; Kim and Hopke, 2007.*

110 **Reply**:

Kim, E., Hopke, P. K., and Qin, Y.: Estimation of organic carbon blank values and error structures of the speciation trends network data for source apportionment, J. Air Waste Manage. Assoc., 55, 1190-1199, doi: 10.1080/10473289.2005.10464705, 2005.

115 Kim, E., and Hopke, P. K.: Comparison between sample-species specific uncertainties and estimated uncertainties for the source apportionment of the speciation trends

network data, Atmos. Environ., 41, 567-575, doi: 10.1016/j.atmosenv.2006.08.023, 2007.

*Lines 278-279: Norris et al., 2014.*

**Reply**:

Norris, G., Duvall, R., Brown, S., Bai, S.: EPA Positive Matrix Factorization (PMF) 5.0 fundamentals and user guide prepared for the US Environmental Protection Agency Office of Research and Development. *Washington, DC*, 2014.

*Line 279: Brown et al., 2015; Paatero et al., 2014; Wang et al., 2017.*

**Reply**:

Brown, S. G., Eberly, S., Paatero, P., and Norris, G. A.: Methods for estimating uncertainty in PMF solutions: Examples with ambient air and water quality data and guidance on reporting PMF results, Sci. Total Environ., 518-519, 626-635, doi: 10.1016/j.scitotenv.2015.01.022, 2015.

Paatero, P., Eberly, S., Brown, S. G., and Norris, G. A.: Methods for estimating uncertainty in factor analytic solutions, Atmos. Meas. Tech., 7, 781-797, doi: 10.5194/amt-7-781-2014, 2014.

Wang, Q., He, X., Huang, X. H. H., Griffith, S. M., Feng, Y., Zhang, T., Zhang, Q., Wu, D., and Yu, J. Z.: Impact of secondary organic aerosol tracers on tracer-based source apportionment of organic carbon and $PM_{2.5}$: A case study in the Pearl River Delta, China, ACS Earth Space Chem., 1, 562-571, doi: 10.1021/acsearthspacechem.7b00088, 2017.

*Line 414: Hjortenkrans et al., 2007.*

**Reply**:

Hjortenkrans, D. S. T., Bergbäck, B. G., and Häggerud, A. V.: Metal emissions from brake linings and tires: Case studies of Stockholm, Sweden 1995/1998 and 2005, Environ. Sci. Technol., 41, 5224-5230, doi: 10.1021/es070198o, 2007.

*Line 416: Thorpe et al., 2008.*

**Reply**: It should be "Thorpe and Harrison, 2008".

Thorpe, A., and Harrison, R. M.: Sources and properties of non-exhaust particulate

150          matter from road traffic: A review, Sci. Total Environ., 400, 270-282, doi:

          10.1016/j.scitotenv.2008.06.007, 2008.

*Line 435: Chang et al., 2016a. Should the "2016a" perhaps be replaced by "2016"?*

**Reply**: Yes, revised accordingly.

155

*Line 520: Wang et al., 2013. There are "Wang et al., 2013a" and "Wang et al., 2013b"*
*in the Reference list.*

**Reply**: The "*Wang et al., 2013*" changed as *"Wang et al., 2013b"* in the text.

160  *Line 555: Karanasiou et al., 2014. There is "Karanasiou, 2014" in the Reference list.*
**Reply**:

"Karanasiou et al., 2012, 2014" in the text should be changed as "Karanasiou et al.,
2012; Karanasiou, 2014".

165  *12. The is no reference made within the text to the following references that are in the*

*Line 25: Replace "at Shanghai" by "at the Shanghai".*

*Line 28: Replace "Elements related" by "Element-related".*

*Line 30: Replace "atmospheric" by "the atmospheric".*

*Line 33: Replace "of elements in" by "of the elements in the".*

190  *Line 34: Replace "in brackets" by "in parentheses".*

*Line 35: Replace "melting" by "smelting".*

*Line 37: Replace "Contributed by exhaust and non-exhaust vehicle emissions," by "The contribution from the exhaust and non-exhaust vehicle emissions, i.e., the".*

*Line 38: Replace "shows strong" by "shows a strong".*

195  *Line 39: Replace "during rush" by "during the rush" and replace "Shipping" by "The shipping".*

*Line 40: Replace "almost" by "are almost".*

*Line 41: Replace "transported from East" by "transported from the East" and replace "fallen" by "falls".*

200  *Line 42: Replace "V/Ni in" by "V/Ni ratios in".*

*Line 43: Replace "K were derived" by "the K was derived".*

*Line 45: Replace "to trace" by "to the trace".*

*Line 47: Replace "of trace" by "of the trace".*

*Line 48: Replace "lower" by "lower the concentration of the".*

205  *Line 50: Replace "in urban" by "in the urban".*

*Line 61: Replace "refers" by "referring".*

*Line 63: Replace "with unique" by "unique".*

*Line 67: Replace "on aerosol" by "on the aerosol".*

*Line 69: Replace "elements components" by "elemental components".*

210  *Line 70: Replace "exerted a" by "exert a".*

*Line 77: Replace "elements species" by "elemental species".*

*Line 87: Replace "of elemental" by "of the elemental".*

*Line 92: Replace "in ambient" by "in the ambient".*

*Line 96: Replace "trace elements" by "trace element".*

215  *Line 102: Replace "for elements" by "for elemental".*

*Line 108: Replace "ambient elements" by "ambient elemental".*

*Line 109: Replace "trace elements" by "trace element".*

*Line 120: Replace "National" by "the National".*

*Line 124: Replace "and Xact" by "and the Xact".*

220  *Line 126: Replace "ambient elements" by "ambient elemental".*

*Line 128: Replace "reactive filter tapes" by "a moving filter tape" and replace "by US"*
*by "by the US".*

*Line 138: Replace "trade, which has" by "trade and has".*

*Line 141: Replace "trace elements" by "trace element".*

225  *Line 147: Replace "elements species" by "elemental species".*

*Line 159: Replace "of trace elements will be investigated to examine if water spay" by*
*"of the trace elements was investigated to examine if water spray".*

*Line 160: Replace "trace elements" by "trace element".*

*Lines 182 and 183: Replace "Pudong" by "the Pudong".*

230  *Line 184: Replace "Pudong" by "The Pudong".*

*Line 185: Replace "For PEMC" by "For the PEMC".*

*Line 188: Replace "indicted in Fig. 1c, PEMC" by "indicated in Fig. 1c, the PEMC"*

*and replace "emissions sources" by "emission sources".*

*Line 193: Replace "PEMC" by "the PEMC".*

*Lines 212-213: Replace "of Xact" by "of the Xact" and replace "on blank Nuclepore" by "on a blank Nuclepore filter".*

*Line 228: Replace "elemental analysis" by "the elemental analysis".*

*Line 233: Replace "using" by "using the".*

*Line 238: Replace "collected by the" by "for the".*

*Line 242: Replace "by Shanghai" by "by the Shanghai".*

*Line 243: Replace "PEMC" by "the PEMC".*

*Line 260: Use superscript for "th" in "kth".*

*Line 268: Replace "elements in" by "elements in the".*

*Line 276: Replace "et al. 1998" by "et al., 1998".*

*Line 284: Replace "with BS" with "with the BS".*

*Line 293: Replace "with similar" by "with a similar".*

*Line 312: Replace "has described" by "has been described".*

*Line 316: Replace "of hourly" by "of the hourly".*

*Line 317: Replace "at PEMC" by "at the PEMC".*

*Line 323: Replace "summaries" by "summary".*

*Line 325: Replace "are also shown" by "are shown".*

*Line 331: Replace "ambient average mass concentrations of elemental" by "the ambient average mass concentrations of the elemental".*

*Line 332: Replace "between detection" by "between the detection".*

*Line 342: Replace "airborne" by "the airborne".*

*Line 345: Replace "on total" by "for the total".*

*Line 350: Replace "concentration as" by "concentration was", replace "which accounting" by "accounting", and replace "of total" by "of the total".*

*Line 363: Replace "although a huge" by "a huge".*

*Line 365: Replace "were given" by "was given".*

*Line 372: Replace "trace elements" by "trace element".*

*Line 374: Replace "an order or two" by "one or two".*

*Line 375: Replace "ranged in the same level as industrialized city" by "were of the same level as in industrialized cities".*

265 *Lines 376 and 377 and within Table 1: Use the same city name, either Gwangju or Kwangju.*

*Line 377: Replace "that of" by "those at".*

*Line 381: Replace "trace elements" by "trace element".*

*Line 382: Replace "elements species" by "elemental species".*

270 *Line 384: Replace "trace elements" by "trace element".*

*Line 390: Replace "of the constant" by "of a constant".*

*Line 392: Replace "explained by" by "obtained with".*

*Line 403: Replace "to factor" by "to the factor" and replace "sources to" by "sources to the".*

275 *Line 408: Replace "the variation" by "the concentration".*

*Line 410: Replace "In urban" by "In the urban".*

*Line 415: Replace "tracers for" by "tracers for a".*

*Lines 417-418: Replace "origins typically" by "origin is typically".*

*Line 425: Replace "assigned to" by "assigned to a".*

280 *Line 432: Replace "of trace" by "of the trace".*

*Line 434: Replace "hours are" by "hours that are".*

*Line 435: Replace "of traffic" by "of the traffic".*

*Line 438: Replace "comes almost" by "come almost".*

*Line 443: Replace "from east" by "from the east".*

285 *Line 444: Replace "from costal" by "from the coastal".*

*Line 447: Replace "Recent" by "A recent".*

*Line 450: Replace "in urban" by "in the urban" and replace "with slightly" by "with slight".*

*Line 458: Replace "during four" by "during the four".*

290 *Line 459: Replace "of trace elements in Shanghai" by "of the trace elements in the*

*Shanghai".*

*Line 460: Replace "its share" by "their share" and replace "in harbor" by "in the harbor".*

*Line 461: Replace "Shanghai and its" by "Shanghai and the".*

295     *Line 463: Replace "in main engines of ships" by "in the main engines of the ships".*

*Line 464: Replace "on air" by "on the air".*

*Line 469: Replace "form of" by "forms of".*

*Line 471: Replace "during roasting" by "during the roasting".*

*Line 475: Replace "to total" by "to the total".*

300     *Line 483: Replace "shows Au" by "shows that Au".*

*Line 484: Replace "that different from Ag" by "which is different from that for Ag".*

*Line 487: Replace "Therefore, element" by "Therefore, the element".*

*Line 491: Replace "the variation" by "the concentration" and replace "and unexpected" by "and an unexpected".*

305     *Line 495: Replace "norther China" by "northern China".*

*Line 502: Replace "from norther China can transport air" by "from northern China transporting air".*

*Line 503: Replace "to Shanghai" by "to the Shanghai".*

*Line 505: Replace "of PM2.5" by "of the PM2.5".*

310     *Line 507: Replace "elements concentrations" by "elemental concentrations".*

*Line 509: Replace "was considered to be originated" by "is considered to originate".*

*Line 511: Replace "of element K in urban Shanghai were" by "of the element K in urban Shanghai was".*

*Line 512: Replace "maybe that" by "may be that".*

315     *Line 514: Replace "has high" by "has a high".*

*Line 519: Replace "contained relatively" by "contained a relatively".*

*Lines 520-521: Replace "in which they detected" by "who observed".*

*Line 522: Replace "i.e." by "i.e.,".*

*Line 523: Replace "ATOFMS" by "the ATOFMS".*

320 *Line 526: Replace "the explained variation" by "the concentration".*

*Line 530: Replace "of world" by "of the world".*

*Line 536: Replace "Since element Cr is reported to be transported over distances by air" by "Since the element Cr is reported to be transported over substantial distances by the air".*

325 *Line 540: Replace "of PM2.5" by "of the PM2.5".*

*Line 550: Replace "particles on the road surface" by "particles".*

*Line 551: Replace "atop tall" by "atop a tall".*

*Line 555: Replace "influence PM" by "influence the PM".*

*Line 556: Replace "of precipitation" by "of the precipitation".*

330 *Line 557: Replace "elements concentration" by "the elemental concentrations".*

*Line 560: Replace "of trace" by "of the trace".*

*Line 561: Replace "The precipitation" by "A precipitation".*

*Lines 563 and 564: Replace "should less" by "should be less".*

*Line 567: Replace "treated as" by "treated as a".*

335 *Line 568: Replace "with the duration" by "with a duration".*

*Line 572: Replace "Fig. S13" by "Fig. S14".*

*Lines 572-573: Replace "scavenge and remove" by "scavenges and removes".*

*Line 574: Replace "should lower" by "should be lower".*

*Line 576: Replace "ambient elements mass" by "the ambient elemental mass".*

340 *Line 578: Replace "lasted" by "which lasted".*

*Line 582: Replace "The variation" by "Variation".*

*Line 585: Replace "decrement rate of Zn" by "decrease of the Zn".*

*Line 588: Replace "were aggregated" by "was aggregated" and replace "Before" by "Before the".*

345 *Lines 589-590: Replace "than that during" by "than those during the".*

*Line 591: Replace "reduce PM2.5" by "reduce the PM2.5" and replace "After precipitation" by "After the precipitation".*

*Line 593: Replace "than that during" by "than during the" and replace "potential" by*

*"a potential".*

350 *Line 595: Replace "as the most" by "as a most" and replace "reflect the" by "reflects the".*

*Line 597: Replace "decrement rate of Zn concentration and" by "decrease of the Zn concentration and the".*

*Line 599: Replace "lower ambient" by "lower the ambient".*

355 *Line 601: Replace "3.3.1" by "3.3.2".*

*Line 610: Replace "source mainly includes" by "sources mainly include".*

*Line 611: Replace "road surface" by "road surfaces".*

*Line 612: Replace "of coal" by "of the coal".*

*Line 613: Replace "and Hg have" by "and Hg having a".*

360 *Line 614: Replace "of coal" by "of the coal".*

*Line 616: Replace "different from traffic-related source, coal" by "different from the traffic-related source, the coal".*

*Line 619: Replace "kept quite" by "remained quite".*

*Line 626: Replace "The variation" by "Variation".*

365 *Line 632: Replace "from Eastern" by "from the Eastern".*

*Line 633: Replace "to urban" by "to the urban".*

*Line 638: Replace "from Eastern China sea and then accumulated in" by "from the Eastern China sea which then accumulated in the".*

*Line 641: Replace "China can" by "China which can".*

370 *Line 643: Replace "in Shanghai" by "in the Shanghai".*

*Line 646: Replace "roses plots" by "roses".*

*Line 648: Replace "bracket" by "parentheses".*

*Line 652: Replace "multi-metal monitor" by "multi-metals monitor".*

*Line 661: Replace "of metal related species comprised" by "of the metal related species*

375 *amounted to".*

*Line 664: Replace "measurement and" by "and".*

*Line 667: Replace "melting" by "smelting".*

*Line 669: Replace "of ambient" by "of the ambient".*

*Line 672: Replace "in urban" by "in the urban".*

380 *Lines 674-675: Replace "sources of emission" by "emission sources".*

**Reply**: Sorry for wasting you so much time. We should avoid most of these mistakes. All the comments you mentioned have been revised one by one. Please refer to our revised MS with track-change model.

385 Comments for the Supplement:

The Supplement would benefit from numbering the pages.

*15. Page 2: The regression data in Table S1 contain too many significant figures. Two significant figures suffice in case the first significant figure is larger or equal than 2, and when the first significant figure equals 1, three significant figures can be used.*

390 **Reply**: In the first round of review, the reviewer suggested "to do the following analyses: Xact vs. glass fiber filters for the 8 filters, Xact vs. cellulose filters for the 8 filters, Xact vs. glass fiber filters for all glass fiber filters, Xact vs. cellulose filters for all cellulose filters, and then perhaps Xact vs. all filters… I propose to add the respective columns to Table S1, and to consequently distinguish the two filter types in the regression

395 analyses". Thank you for your understanding.

*16. Page 3, first line: Replace "Text S2" by "Table S2" and replace "precipitation chosen" by "precipitation events chosen".*

*Page 4, line 1: Replace "Text 1" by "Text S1".*

400 *Page 4, line 13: Replace "Table below" by "Table S3" and replace "95 of BS" by "95 of the BS".*

*Page 4, line 14: Replace "50 of BS" by "50 of the BS".*

*Page 4, line 15: Replace "and other" by "and the other" and replace "95 of BS" by "95 of the BS".*

405 *Page 4, line 6 from bottom: Replace "perform PMF" by "performs PMF".*

*Page 4, line 3 from bottom: Replace "Table below" by "Table S4".*

*Page 5, line 1 below Table S4: Replace "of BS and DISP method" by "of the BS and DISP methods".*

*Page 5, line 5 below Table S4: Replace "for PMF" by "for the PMF".*

410 *Page 5, line 6 below Table S4: Replace "When factor" by "When the factor".*

*Page 5, line 7 below Table S4: Replace "than from 4- to 5-factor" by "than is the case when going from the 4- to the 5-factor" and replace "However, a" by "The".*

*Page 5, line 8 below Table S4: Replace "predetermined" by "selected".*

*Page 6, line 10: Replace "781" by "781-797".*

415 *Page 7, line 1 of figure caption: Replace "Figure 2. Seasonal variations" by "Figure S2. Seasonal variation".*

*Page 15, line 3 of caption of Figure S10: Replace "Below the diagonal of the diagonal" by "Below the diagonal".*

*Page 18, caption of Figure S13: Replace "The distribution of precipitation" by "Distribution of precipitation events".*

420 *Page 19, line 1 of caption of Figure S14: Replace "The variation" by "Variation".*

**Reply**: We corrected accordingly in the revised MS.

*17. Page 9: I do not see any wind direction information in Figure S4. Should ", and*

425 *wind directions" not be deleted in the figure caption?*

**Reply**: Followed the suggestion by the reviewer in the first round of review, we decided to delete the wind direction information. We also think that it is very strange to plot the normalized data of metal concentration and wind direction in one figure.

430

435

Anonymous Referee #6

1.  The paper *"First long-term and near real-time measurement of trace elements in China's urban atmosphere: temporal variability, source apportionment, and precipitation effect"* is based on a huge and interesting data set. Nevertheless, the paper still includes some points that need revision. Briefly, here is a list of the issues that authors should address for the publication of the paper:

- To the best of my knowledge, Xact can also quantify Al, S, Cl, and Ti: why were these elements not taken into account?

**Reply**: Yes, Xact can also quantify Al, S, Cl, and Ti. However, firstly, these four elements have the highest detection limits and their data quality are generally not as good as other elements. Secondly, the display interface of Xact (old version) has limited space to show all the measured trace elements, and the Xact multi-metals monitor used in our study was routinely deployed. For the consideration of data quality assurance (and convenience), we (and also most of other Xact users) decided to remove the data of Al, S, Cl, and Ti.

2.  - Still on Xact: what do you mean with *"reactive filter tapes"* (line 128)? Referees had already asked this question.

**Reply**: Sorry for making such confusion to you. The "reactive filter tapes" means "moving filter tapes", and we've corrected it in the revised MS as suggested by the handling editor of our MS.

3.  - There is quite some confusion between V/Ni and Ni/V ratios: please report only V/Ni ratio values, as this is the main way to report these data. Please revise the abstract (line 41) and figure 6.

**Reply**: Sorry for our mistake. We've corrected the text and figure in our revised MS.

4.  - Lines 347-352: the issue of oxides versus reconstructed mass is addressed in a quite simplistic way: could the authors explain the formulas they used for the

465   *calculation of the oxides and, mainly, how was the data for total PM2.5 mass*
    *obtained? The cited reference helps under the methodological point of view, but it*
    *would be helpful to know some more specific details (how did the authors calculate*
    *EC/OC, sulphate, nitrate etc. for the reconstruction of the mass).'*

  **Reply**: Our sampling site is responsible for publicly releasing hourly air quality data of
470  $PM_{10}$, $PM_{2.5}$, and other criteria pollutants (CO, $SO_2$, $NO_x$ and $O_3$) in Shanghai. The
  hourly mass concentrations of $PM_{2.5}$ were measured by a particulate monitor (Thermo,
  FH62C-14). Therefore, there is no need to perform the calculation of $PM_{2.5}$ mass
  closure in our study. We've added this information in the section of 2.1.3, i.e.,
  "Auxiliary measurements".

475

    *5. - Line 450: which is the V/Ni value of the PMF factor? The factor is obviously*
    *characterized by the same ratio throughout the year; there is an unclear mix among*
    *PMF factor data and raw chemical data.*

  **Reply**: In the PMF analysis, we found that V (100%) and Ni (74%) come almost
480  exclusively from factor 2, while factor 2 contributes to less than 10% of any other
  elemental species. Therefore, the evolution of PMF factor 2, to a large extent, reflects
  the variation raw chemical data of shipping-related emissions of trace elements.

    *6. - Figure 7, CPF and BPP analysis: I suggest the authors to correlate wind data with*
485    *the factors/sources, while they did it with elements. This would give more powerful*
    *information on sources, even though a single element could have contributions from*
    *different sources.*

  **Reply**: We agree with the reviewer that the CPF and BPP analysis could give powerful
  information on sources. However, when we have the PMF-derived **quantitative** results
490  of the sources, then the CPF and BPP analysis should be an auxiliary tool. Also, we
  agree with the reviewer that a single element could have contributions from different
  sources. However, the PMF-derived quantitative results of the elemental sources in our
  study are interpretable and convincing. Therefore, we have no urgent need to perform

the CPF and BPP analysis for every factor/element. Nevertheless, the CPF and BPP

495    analysis have been applied for Au and Hg, and Cr in our study.

7.   - *Section 3.3, "Precipitation effect". The topic is interesting, but authors should*
      *comment their results somehow more critically: e.g., how do they explain the fact*
      *that Au and Cr increase during the rainfall? I find this result quite strange... Further,*

500      *no comment is given on the ferrous metal smelting pattern (fig. 11 e). Last,*
      *comparing the effect of a rainfall to the expected one for street-washing appear*
      *quite simplistic, as in the first case the whole "air column" is washed, while with*
      *street washing only the street surface is cleaned.*

**Reply**: (1) "how do they explain the fact that Au and Cr increase during the rainfall".

505    The discussion of precipitation effect in the MS was divided into two parts, i.e., "3.3.1
Change of mass concentration by species" and "3.3.2 Change of mass concentration by
sources". It is not always easy to explain the increase of trace elements like Au and Cr
during the rainfall in section 3.3.1 alone. However, when we look at the PMF section,
the predominant elements found in factor 3 (nonferrous metal smelting sector) were Au

510    (100%), Cd (65%), and Ag (63%) with 37% of Hg. Meanwhile, factor 5 (ferrous metal
smelting sector) was distinguished by high levels of Cr, Mn, and Zn, which represent
100%, 56%, and 52% of the explained concentration, respectively. Therefore, Au and
Cr were almost exclusively emitted from nonferrous metal smelting sector and ferrous
metal smelting sector, respectively. In the section 3.3.2, different from the sectors of

515    traffic, shipping, and coal combustion, we found that the changes of mass
concentrations of trace elements from nonferrous metal smelting sector and ferrous
metal smelting sector were very limited during rainfall. This is generally in accordance
with the section 3.3.1 that "Au and Cr increase during the rainfall".

(2) "comparing the effect of a rainfall to the expected one for street-washing appear

520    quite simplistic, as in the first case the whole "air column" is washed, while with street
washing only the street surface is cleaned."

As we described in the MS, water spray (from sprinkler on road or atop a tall building,

**i.e., water-fog cannon**) in our study including but not limited to street washing. Indeed, in China, some radical measures like "water-fog cannon" (see Yu, S.: Water spray geoengineering to clean air pollution for mitigating haze in China's cities, Environ. Chem. Lett., 12, 109-116, doi: 10.1007/s10311-013-0444-0, 2014) and "smog cleaning tower" (see https://www.nature.com/articles/d41586-018-02704-9#menu) have been proposed to curb air pollution.

We completely agree with the reviewer that during a rainfall event, "the whole "air column" is washed. And we think that the scavenging effect of rainfall on the ambient trace elements should be much stronger than that of water-spray activities. So our logic is this: If there is no scavenging effect of precipitation on a given source or species of trace element, then we can conclude that it is useless to take water spray as measure to lower the ambient trace elements. If there is a strong scavenging effect of precipitation on a given source or species of trace element, then we can conclude that water spray may be a measure to lower the ambient trace elements. Our results show that the precipitation has limited scavenging effect on the sources of shipping, nonferrous metal smelting, and ferrous metal smelting. However, the ambient concentrations of trace elements from traffic and coal combustion (accounting for 67% of the total mass) experienced a dramatic decrease during rainfall. Therefore, in general, we concluded that water spray may be a way to lower the ambient trace elemental concentration.

8. *- Line 555: in the reference list one more paper by Karanasiou et al. is reported (published in 2011).*

**Reply**: The paper "Karanasiou et al., 2011" was correctly cited.

9. *- Supplementary material, table S1. Actually, Further, with respect to Fe: could the authors check the regression lines? I find strange that the intercept for all glass filters and all cellulose filters are negative but, then, the intercept for all the filters is positive. Last, what do "*" and "**" stand for?*

**Reply**: (1) "the inter-comparison between ICP-MS and Xact gives not excellent results

for some elements (e.g., As, Cd, Ba, Pb, for which a slope of roughly 2 is observed): this issue should be discussed in more detail."

Our initial thought about the explanation was that the filters we used are not as good as Teflon filter to collect the ambient trace elements. Prof. Mei Zheng from Peking University recently also performed a method comparison between Xact and ICP-MS (using Teflon filters) in northern China (see table below, for reviewing purpose use only). Unfortunately, except for Pb, the average concentrations of elements like Cd, Ba, and As measured by a Xact were also over two times higher than that measured by ICP-MS. The frustrated thing is that we failed to find another explanation at the current stage.

| | XACT ( N=1158) | | | ICP-MS (N=62) | | |
|---|---|---|---|---|---|---|
| | Mean | STD | Median | Mean | STD | Median |
| | ng m$^{-3}$ | | | ng m$^{-3}$ | | |
| K | 2052.71 | 2022.7 | 1562.5 | 1183.222 | 1069.39 | 887.68 |
| Fe | 534.67 | 263.11 | 513.1 | 436.82 | 235.91 | 408.12 |
| Ca | 481.41 | 438.24 | 358.1 | 642.48 | 460.71 | 494.83 |
| Zn | 434.25 | 547.15 | 281.3 | 274.532 | 175.675 | 238.705 |
| Pb | 121.95 | 96.35 | 95.37 | 97.196 | 57.1 | 83.189 |
| Mn | 31.96 | 26.16 | 26.3 | 25.45 | 12.558 | 22.096 |
| Cu | 31.12 | 37.65 | 20.22 | 24.34 | 19.44 | 19.05 |
| Cd | 24.4 | 9.6 | 23.93 | 2.948 | 1.991 | 2.58 |
| Ba | 23.32 | 12.37 | 21.42 | 11.131 | 7.068 | 9.65 |
| As | 12.81 | 16.35 | 9.43 | 5.629 | 8.088 | 2.365 |
| Ag | 10.29 | 6.91 | 6.12 | | | |
| Se | 6.13 | 4.79 | 5.43 | 5.376 | 3.917 | 4.465 |
| Hg | 2.77 | 1.64 | 2.5 | | | |
| Au | 2.57 | 1.94 | 2.22 | | | |
| Cr | 2.05 | 2.26 | 1.47 | 6.102 | 5.333 | 4.385 |
| Ni | 1.77 | 1.68 | 1.47 | 1.92 | 1.921 | 1.14 |
| V | 0.8 | 1.03 | 0.41 | 6.93 | 10.62 | 3.63 |

(2) "with respect to Fe: could the authors check the regression lines? I find strange that the intercept for all glass filters and all cellulose filters are negative but, then, the intercept for all the filters is positive. Last, what do "*" and "**" stand for? "

After double check, our regression lines are correct. The reason of the "strange" positive intercept for all the filters is because that an abnormal value of Fe (675.0 ng m$^{-3}$) collected by glass filter was excluded, which is indicated as "*". The original formula

of "all glass filters vs. Xact" and "all filters vs. Xact" are "y=0.77x+119.3 $R^2$ =0.38 n=11" and "y=0.95x+22.28 $R^2$ =0.70 n=31", respectively. Similarly, "**" stands for the deletion of two abnormal values of As (63.0 and 77.7 ng m$^{-3}$) collected by cellulose filters.

*10. - Supplementary material, figure S4. Wind directions are not reported: please add the panel or change the caption.*

**Reply**: Followed the suggestion by the reviewer in the first round of review, we decided to delete the wind direction information. We also think that it is very strange to plot the normalized data of trace elemental concentration and wind direction in one figure.

Anonymous Referee #2

General comments

*1. The manuscript has been significantly altered by basically replacing the hierarchical clustering method by Positive Matrix Factorization (PMF), and by adding information on precipitation effects on metals concentrations. PMF yields 5 emission sources for Shanghai which are well explained by the measured elemental data, and the high time resolution allows for a strong argument for a traffic-related factor. Precipitation effects do show a general trend in reducing the elemental concentrations, however the picture is complicated by other effects playing a role, such as wind/air mass changes. There are still a few minor changes required before the paper can be published. Specific comments*

*The comparison between filters and Xact has been extended in Table S1, but the description (lines 234-239) is not consistent with the new data. The background values have been dropped from the table, as well as the comparison between the filter types. The quoted values on line 237 refer to the very first version of the table. This should be updated. I still do not buy the explanation of the Cr and Ba slopes depending on the background values. If the background values in the original version of Table S1 were correct, than all cellulose acetate filter background values were smaller than the glass filter background values, in contradiction to the statement. Perhaps the authors leave out that part of the sentence (lines 238-239). Table S1 needs more information in the caption, e.g. explain the asterisks (\*, \*\*) and the n values.*

**Reply**: Sorry for our carefulness, we updated the results in the revised MS. We decided to delete "the slope values of Cr (1.9) and Ba (2.6) were higher than those of other species, this can be explained by higher background values of Cr and Ba collected by the cellulose acetate filters". "\*" indicates that an abnormal value of Fe (675.0 ng m$^{-3}$) collected by glass filter was excluded. "\*\*" stands for the deletion of two abnormal values of As (63.0 and 77.7 ng m$^{-3}$) collected by cellulose filters.

625  2. *In Section 2.1.1. only winter is described. I recommend to add the sequence from*
*the first revision (with my additions): "Winters are chilly and damp, with*
*northwesterly winds from Siberia sometimes causing nighttime temperatures to*
*drop below freezing. In summer, airflow carries moist air from the Pacific Ocean to*
*mainland China, which will also bring the main precipitation. The city is also*
630  *susceptible to typhoons in summer and the beginning of autumn."*
   **Reply**: Thanks, we've added these sentences in the revised MS.

   3. *Technical corrections Table 1, L376, L377 make city names consistent L572*
*Fig. S14 (not S13)*
635  **Reply**: Revised accordingly.

   4. *In supplement*
   *Table S1:  explain 'n' and the asterisks in the table*
   **Reply**: "n" represents the number of paired samples.
640

   5. *Table S2:  Replace 'Text' with 'Table', and insert 'events ' after 'precipitation'*
   **Reply**: Revised accordingly.

   6. *Caption of Fig. S10: L3 delete 'of the diagonal'*
645  **Reply**: Revised accordingly.

[revised manuscript text omitted]

---

## Author Response (AR4)

Dear Prof. Maenhaut,

First of all, we would like to clarify that we didn't want to deliberately avoid to address your comments regarding "significant figures". We thought the "figure" means "graph". Sorry for our misunderstanding.

The revised MS (P9-50) and supplement (P51-69) are attached below the reply letter.

Comment 1: *The authors could have addressed some of the comments of the two Anonymous Referees and of myself in a better way. The comments of Anonymous Referee #6 about "Precipitation effect" should be properly taken into consideration, in particular, "how do the authors explain that Au and Cr increase during the rainfall"; consequently, revision is needed within section 3.3.1.*

Reply:

[Figure]

Although we cannot address this comment in a quantitative manner, we have a strong believe that the height of emission, the location of emitter, the timing of emission are three major factors influencing the "precipitation effects".

Firstly, we would like to give you a typical example-**Ca**. In our study, almost all Ca we measured can be explained by the contribution of road fugitive dust blown by traffic flow. In other words, the ambient Ca concentrations we measured were largely originated from road surface with a emission height of **0** m. Therefore, a wet surface after rainfall can be expected to dramatically reduce the amount of fugitive dust.

Besides, traffic is ubiquitous throughout Shanghai city, and the sampling site in our study is also surrounded by roads. Moreover, despite a much lower traffic flow during nighttime, fugitive dust can be expected to be continuously generated by on-road traffic. Consequently, Ca concentrations experienced the biggest drop during rainfall in our study.

As to **Au** and **Cr**, they were almost entirely originated from nonferrous metal smelting and ferrous metal smelting in our study, respectively.

(1) the height of emission: Different from road fugitive dust, atmospheric pollutants exhausted from nonferrous metal smelting and ferrous metal smelting are typically emitted through elevated chimney.

(2) the location of emitter: Nonferrous metal smelting and ferrous metal smelting activities in Shanghai are concentrated in suburban and rural areas, which are far away from our sampling site.

the timing of emission: Nonferrous metal smelting and ferrous metal smelting activities generally existed during daytime of working days. Therefore, even if there is a rainfall event during daytime of a working day, the ambient Au and Cr concentrations can be expected to higher than that of non-rainy hours.

We've revised section 3.3.1 with a focus on the explanation of a increase of Au and Cr during the rainfall (line 929-937).

Comment 2: *Also my comment about significant figures was definitely not properly*

*addressed.*

Reply: All your comments about significant figures in our revised MS have been corrected. As we mentioned above, we didn't want to deliberately avoid to address your comments regarding "significant figures". We thought the "figure" means "graph".

Comment 3: *Furthermore, the authors failed to address several corrections, which I listed in my previous report. They are listed again. Besides, I have a few new corrections, including regarding the wording and grammar in the new sections of text. Consequently, additional revision is needed before this manuscript can be accepted for*

*publication in ACP.*

Reply: Thank you very much for your careful revision. We've addressed all the corrections in our revised MS.

*Main comments for the Main text:*

Comment 4: *1. Page 13, within Figure 2 (and on various other occasions in the manuscript, e.g., in Table 1, data for this study; in line 339; within Figure 3; in line 409; in line 427): The concentration data contain too many significant figures (digits). Two significant figures suffice in case the first significant figure is larger than (or equal to) 2, and when the first significant figure equals 1, three significant figures can be*

*used. For example, in Table 1 the authors' data should become as follows: Ba 24, Ca 192, Fe 410, K 390, Mn 32, Pb 27, Si 640, Zn 120.*

Reply: We've thoroughly addressed your concern regarding significant figures in our revised MS. Particularly, we re-plotted Figure 2 as shown below.

[Figure]

Comment 5: *2. There are still problems with the References.*

*The following references to which is referred within the text are not in the Reference*

*list:*

*Line 261: Paatero and Tapper, 1994.*

Reply: Paatero, P. and Tapper, U.: Positive matrix factorization: A non-negative factor model with optimal utilization of error estimates of data values, Environmetrics, 5, 111-126, doi: 10.1002/env.3170050203, 1994.

Comment 6: *Line 281: Polissar et al., 1998.*

Reply: Polissar, V., Hopke, P., Paatero, P., Malm, W., and Sisler, J: Atmospheric aerosol over Alaska: 2. Elemental composition and sources, J. Geophys. Res., 103, 19045-19057, doi: 10.1029/98JD01212, 1998.

Comment 7: *There is no reference made within the text to the following reference that*
*is in the Reference list:*

*Pentti and Unto, 1994.*

Reply: Pentti and Unto are the fist authors. The correct reference is "Paatero, P. and Tapper, U.: Positive matrix factorization: A non-negative factor model with optimal utilization of error estimates of data values, Environmetrics, 5, 111-126,
doi: 10.1002/env.3170050203, 1994."

Comment 8: *The references in the Reference list are not always in the appropriate order:*

*"Karanasiou, 2014" should come before "Karanasiou et al., 2011".*
*"Liu, S. et al., 2014" should come before "Liu, Z. et al., 2015".*

*Furthermore:*

*Line 960: Replace "L., and" by "L. and".*

Reply: Revised accordingly.

Comment 9: *Further comments for the Main text:*

*Line 28: Replace "Elements-related" by "Element-related".*

*Line 48: Replace "of trace" by "of the trace".*

*Line 59: Replace "Pope III et al., 2002; Pope III et al., 2009" by "Pope III et al., 2002, 2009".*

*Line 110: Replace "trace elements" by "trace element".*

*Line 115: Replace "Visser et al., 2015b; Visser et al., 2015a" by " Visser et al., 2015b, 2015a).*

*Line 142: Replace "trace elements" by "trace element".*

*Line 174: Replace "airflow" by "the airflow".*

*Line 175: Replace "which will" by "and it will".*

*Line 187: Replace "Pudong New" by "the Pudong New".*

*Line 237: Replace "using" by "using the".*

*Line 238: Replace "different methods" by "different measurement methods".*

*Line 239: Replace "to measure and different filters (glass filter vs. cellulose filter) to*

*collect trace elements" by "and different collection substrates (glass filter vs. cellulose*

*filter)".*

*Line 240: Replace "were reported in Table S2. In general, elements" by "are reported*

*in Table S1. In general, the data for elements" and replace "Au were" by "Au".*

*Line 446: Replace "costal port" by "coastal port".*

*Line 457: Replace "20014" by "2014".*

*Line 568: Replace "should less" by "should be less".*

*Line 598: Replace "than that during" by "than during".*

*Line 603: Replace "and rainfall" by "and the rainfall".*

*Line 666: Replace "accounted to" by "amounted to".*

Reply: Done. Thank you so much for your valuable time to correct our errors.

Comments for the Supplement:

Comment 10: *The Supplement would benefit from numbering the pages.*

Reply: Done.

Comment 11: *Page 2: The regression data in Table S1 contain too many significant figures (digits). Two significant figures suffice in case the first significant figure is larger than (or equal to) 2, and when the first significant figure equals 1, three significant figures can be used.*

Reply: Revised accordingly.

| Species | 8 glass filters vs. Xact | all glass filters vs. Xact | 8 cellulose filters vs. Xact | all cellulose filters vs. Xact | all 48 filters vs. Xact |
|---------|--------------------------|----------------------------|------------------------------|--------------------------------|--------------------------|
| K | y=1.06x+135 R²=0.85 n=7 | y=1.07x+72 R² = 0.87 n=21 | y=1.03x+121 R²=0.96 n=8 | y=1.13x+131 R² = 0.91 n=25 | y=1.08x+110 R² = 0.86 n=46 |
| Cr | y=1.13x+0.51 R²=0.92 n=3 | y=1.77x-17 R²=0.75 n=8 | y=1.69x-3.9 R²=0.65 n=6 | y=1.92x-0.54 R²=0.58 n=21 | y=1.83x-0.48 R²=0.49 n=29 |
| Mn | y=0.87x-5.1 R²=0.92 n=4 | y=0.67x+12 R²=0.52 n=18 | y=0.85x-5.9 R²=0.87 n=8 | y=0.89x+9.5 R²=0.66 n=25 | y=0.79x+10.7 R²=0.59 n=43 |
| Fe | y=2.0x-257 R² =0.99 n=3 | y=1.67x-97 R² = 0.95 n=10* | y=0.98x-21 R² =0.81 n=6 | y=1.01x-22 R² = 0.82 n=20 | y=1.03x+6.5 R² = 0.78 n=30* |
| Ni | y=1.62x-6.9 R²=0.32 n=6 | y=2.1x-13 R² =0.53 n=20 | y=1.45x-6.1 R²=0.54 n=7 | y=1.09x+3.7 R² =0.55 n=23 | y=1.32x+0.29 R² =0.49 n=43 |
| Cu | y=1.97x -3.3 R²=0.71 n=5 | y=0.97x+5.6 R²=0.40 n=19 | y=0.89x+5.0 R²=0.71 n=8 | y=1.12+5.1 R²=0.61 n=25 | y=1.10+5.0 R²=0.57 n=44 |
| As | n=2 | y=2.1x+0.19 R²=0.48 n=5 | y=2.2x-42 R²=0.54 n=3 | y=2.3x-19 R²=0.49 n=4** | y=2.1x-16 R²=0.36 n=9** |
| Cd | y=2.3x+1.04 R² = 0.90 n=7 | y=1.99x+9.0 R²=0.82 n=21 | y=2.1x+5.0 R² = 0.80 n=8 | y=1.97x+6.7 R²=0.73 n=24 | y=1.97x+8.1 R²=0.76 n=45 |
| Ba | y=2.9x-0.12 R²=0.77 n=7 | y=3.1x+0.04 R²=0.78 n=21 | y=2.5x-0.18 R²=0.82 n=8 | y=2.4x+0.29 R² =0.78 n=24 | y=2.7x+0.20 R² =0.81 n=45 |
| Au | y=0.97x+3.7 R²=0.55 n=7 | y=1.25x+2.0 R²=0.72 n=21 | y=1.18x+1.77 R²=0.82 n=8 | y=1.15x+1.50 R²=0.84 n=23 | y=1.17x+1.81 R²=0.77 n=44 |
| Pb | y=2.7x-0.12 R²=0.68 n=7 | y=2.1x+0.11 R²=0.54 n=19 | y=2.4x-0.10 R²=0.85 n=8 | y=1.74x+0.04 R²=0.64 n=25 | y=1.80x+0.09 R²=0.59 n=44 |

Comment 12: *Page 5, line 1 below Table S4: Replace "DISP method" by "DISP methods".*

*Page 5, lines 5-6 below Table S4: Replace "Figure below" by "Figure S1".*

*Page 5, line 7 below Table S4: Replace "from 4- to 5-factor" by "from the 4- to the 5-*

*factor".*

*Page 5, line 8 below Table S4: Replace "predetermined" by "selected".*

*Page 6, line 10: Replace "781-791" by "781-797".*

*Page 18, caption of Figure S13: Replace "The distribution of precipitation" by "Distribution of precipitation events".*

*Page 19, line 1 of caption of Figure S14: Replace "Figure 14" by "Figure S14" and*

   *replace "The variation" by "Variation".*

   Reply: Revised accordingly.

[revised manuscript text omitted]
. As depicted in Fig. 3, Au and Cr were almost entirely originated from nonferrous metal smelting and ferrous metal smelting, respectively. Different from road fugitive dust, atmospheric pollutants exhausted from nonferrous metal smelting and ferrous metal smelting are typically emitted through elevated chimney. Besides, these activities are concentrated in the suburban and rural areas of Shanghai, which are far away from our sampling site (Fig. 1). Moreover, metal smelting activities generally existed during daytime of working days. Therefore, even if there is a rainfall event during daytime of a working day, the ambient Au and Cr concentrations can be expected to higher than that of non-rainy hours. 
[revised manuscript text omitted]
**. Inter-comparison between ICP-MS and Xact for trace elements ("n" represents the number of paired samples).

| Species | 8 glass filters vs. Xact | all glass filters vs. Xact | 8 cellulose filters vs. Xact | all cellulose filters vs. Xact | all 48 filters vs. Xact |
|---|---|---|---|---|---|
| **K** | y=1.06x+135 R²=0.85 n=7 | y=1.07x+72 R² = 0.87 n=21 | y=1.03x+121 R²=0.96 n=8 | y=1.13x+131 R² = 0.91 n=25 | y=1.08x+110 R² = 0.86 n=46 |
| **Cr** | y=1.13x+0.51 R²=0.92 n=3 | y=1.77x-17 R²=0.75 n=8 | y=1.69x-3.9 R²=0.65 n=6 | y=1.92x-0.54 R²=0.58 n=21 | y=1.83x-0.48 R²=0.49 n=29 |
| **Mn** | y=0.87x-5.1 R²=0.92 n=4 | y=0.67x+12 R²=0.52 n=18 | y=0.85x-5.9 R²=0.87 n=8 | y=0.89x+9.5 R²=0.66 n=25 | y=0.79x+10.7 R²=0.59 n=43 |
| **Fe** | y=2.0x-257 R² =0.99 n=3 | y=1.67x-97 R² = 0.95 n=10* | y=0.98x-21 R² =0.81 n=6 | y=1.01x-22 R² = 0.82 n=20 | y=1.03x+6.5 R² = 0.78 n=30* |
| **Ni** | y=1.62x-6.9 R²=0.32 n=6 | y=2.1x-13 R² =0.53 n=20 | y=1.45x-6.1 R²=0.54 n=7 | y=1.09x+3.7 R² =0.55 n=23 | y=1.32x+0.29 R² =0.49 n=43 |
| **Cu** | y=1.97x -3.3 R²=0.71 n=5 | y=0.97x+5.6 R²=0.40 n=19 | y=0.89x+5.0 R²=0.71 n=8 | y=1.12+5.1 R²=0.61 n=25 | y=1.10+5.0 R²=0.57 n=44 |
| **As** | n=2 | y=2.1x+0.19 R²=0.48 n=5 | y=2.2x-42 R²=0.54 n=3 | y=2.3x-19 R²=0.49 n=4** | y=2.1x-16 R²=0.36 n=9** |
| **Cd** | y=2.3x+1.04 R² = 0.90 n=7 | y=1.99x+9.0 R²=0.82 n=21 | y=2.1x+5.0 R² = 0.80 n=8 | y=1.97x+6.7 R²=0.73 n=24 | y=1.97x+8.1 R²=0.76 n=45 |
| **Ba** | y=2.9x-0.12 R²=0.77 n=7 | y=3.1x+0.04 R²=0.78 n=21 | y=2.5x-0.18 R²=0.82 n=8 | y=2.4x+0.29 R² =0.78 n=24 | y=2.7x+0.20 R² =0.81 n=45 |
| **Au** | y=0.97x+3.7 R²=0.55 n=7 | y=1.25x+2.0 R²=0.72 n=21 | y=1.18x+1.77 R²=0.82 n=8 | y=1.15x+1.50 R²=0.84 n=23 | y=1.17x+1.81 R²=0.77 n=44 |
| **Pb** | y=2.7x-0.12 R²=0.68 n=7 | y=2.1x+0.11 R²=0.54 n=19 | y=2.4x-0.10 R²=0.85 n=8 | y=1.74x+0.04 R²=0.64 n=25 | y=1.80x+0.09 R²=0.59 n=44 |

Note: "*" indicates that an abnormal value of Fe (675 ng m⁻³) collected by glass filter was excluded. "**" stands for the deletion of two abnormal values of As (63 and 78 ng m⁻³) collected by cellulose filters.

**Table S2. General information of the 12 precipitation events chosen in this study.**

| precipitation event ID | start | ending | accumulated rainfall amount (mm) |
|---|---|---|---|
| 1 | 01-06 17:00 | 01-07 21:00 | 40.7 |
| 2 | 03-08 06:00 | 04-06 05:00 | 52.9 |
| 3 | 04-06 06:00 | 04-07 09:00 | 50.8 |
| 4 | 04-20 15:00 | 04-21 09:00 | 26.4 |
| 5 | 06-12 03:00 | 06-12 23:00 | 74 |
| 6 | 07-02 17:00 | 07-02 23:00 | 35.9 |
| 7 | 09-07 03:00 | 09-07 17:00 | 33.4 |
| 8 | 09-14 03:00 | 09-17 02:00 | 192.4 |
| 9 | 09-29 07:00 | 09-29 23:00 | 37.3 |
| 10 | 10-07 21:00 | 10-08 12:00 | 33.2 |
| 11 | 10-21 01:00 | 10-23 07:00 | 217.5 |
| 12 | 12-25 09:00 | 12-26 21:00 | 44.4 |

**Text S1. PMF uncertainty analysis and factor number determination**

In EPA PMF 5.0, bootstrapping (BS), displacement (DISP), and bootstrapping enhanced with DISP (BS-DISP) are used to estimate the results uncertainty. The methods were introduced in detail by Norris et al. (2014), Brown et al. (2015) and Paatero et al. (2014), and have been applied in source apportionment of $PM_{2.5}$

components (Liu et al., 2017; Wang et al., 2017).

BS analysis involves resampling from the original input data set, performing PMF analysis with resampled data set (bootstrapped solution), and the comparison of factor contributions between base-case and bootstrapped solutions. In this work, 100 BS runs in total were performed, and an *r* value of 0.8 was set to map the bootstrapped factor contributions with those of the base-case. The mapping of each BS factor contribution with the corresponding base-case factor contribution (5-factor solution) is shown in Table S3. All factors are mapped in more than 95 of the BS runs with five factors. If we select a factor number of six, one of the six factors is mapped in only 50 of the BS runs, and the other 5 factors are mapped in more than 95 of the BS runs. These results suggest a maximum factor number of five. Too many factors will split one source into multiple uninterpretable factors.

**Table S3. Summary of error estimation diagnostics from BS.**

| BS Mapping (r ≥0.8) | traffic | shipping | nonferrous metal smelting | coal combustion | ferrous metal smelting | Unmapped |
|---|---|---|---|---|---|---|
| traffic | 98 | 0 | 0 | 0 | 0 | 2 |
| shipping | 0 | 100 | 0 | 0 | 0 | 0 |
| nonferrous metal smelting | 0 | 0 | 100 | 0 | 0 | 0 |
| coal combustion | 0 | 0 | 0 | 100 | 0 | 0 |
| ferrous metal smelting | 0 | 0 | 0 | 0 | 97 | 3 |

DISP adjusts each species in the factor profile up and down one by one, and then performs PMF runs to obtain a change of Q (dQ, $Q_{displaced\ run} - Q_{base\ run}$) less than the selected maximum allowable dQ$^{max}$ (4, 8, 15, 25). For each dQ$^{max}$ value, DISP is conducted and the intervals (minimum and maximum source profile values) are summarized for each species in each factor. The DISP output is shown in Table S4. The error code is 0, meaning no error. The largest observed drop of Q during DISP is also 0. No factor swap occurs for the smallest dQ$^{max}$, suggesting a stable and robust

PMF solution, and there should be no rotational ambiguity.

**Table S4. Summary of error estimation diagnostics from DISP.**

| | traffic | shipping | nonferrous metal smelting | coal combustion | ferrous metal smelting |
|---|---|---|---|---|---|
| Error Code: | 0 | | | | |
| Largest Decrease in Q: | 0 | | | | |
| %dQ: | 0 | | | | |
| Swaps by Factor: | | | | | |
| dQ$^{max}$=4 | 0 | 0 | 0 | 0 | 0 |
| dQ$^{max}$=8 | 0 | 0 | 0 | 0 | 0 |
| dQ$^{max}$=15 | 0 | 0 | 0 | 0 | 0 |
| dQ$^{max}$=25 | 0 | 0 | 0 | 0 | 0 |

BS-DISP is a combination of the BS and DISP methods, where displacement occurs in source profiles derived from each resampled data set. It estimates the errors associated with both random and rotational ambiguity. Due to the huge data set in this work, (7519

× 18), BS-DISP analysis is time consuming and not conducted.

Q/Q$_{exp}$ was calculated for the PMF solutions with factor numbers from 3 to 10 (Fig. S1). When the factor number reaches 9, the Q/Q$_{exp}$ will change less substantially (6.9%) than is the case when going from the 4- to the 5-factor solution (18.9%). However, a 5-factor solution was predetermined due to the most interpretable and stable results.

[Figure]

**Figure S1**. Change of $Q/Q_{exp}$ value from 3-factor to 10-factor solution.

[Figure]

**Figure S2.** Seasonal variation of mass concentrations for 18 trace elements measured
in Shanghai between March 2016 and February 2017. The gray line indicates one standard deviation. Four seasons in Shanghai were defined as follows: March-May as spring, June-August as summer, September- November as fall, and December and January-February as winter.

[Figure]

**Figure S3.** Weekly diurnal (**a**), diurnal (**b**), monthly (**c**), and weekly (**d**) variations of normalized Cd and Ag concentrations in Shanghai.

[Figure]

**Figure S4.** Weekly diurnal (**a**), diurnal (**b**), monthly (**c**), and weekly (**d**) variations of normalized V and Ni concentrations in Shanghai.

[Figure]

**Figure S5.** Weekly diurnal (**a**), diurnal (**b**), monthly (**c**), and weekly (**d**) variations of normalized Si, Ca, Fe, and Ba concentrations in Shanghai.

[Figure]

**Figure S6.** Weekly diurnal (**a**), diurnal (**b**), monthly (**c**), and weekly (**d**) variations of
normalized Mn and Zn concentrations in Shanghai.

[Figure]

**Figure S7.** Weekly diurnal (**a**), diurnal (**b**), monthly (**c**), and weekly (**d**) variations of
normalized Cu, K, Se, As, and Pb concentrations in Shanghai.

[Figure]

**Figure S8.** Weekly diurnal (**a**), diurnal (**b**), monthly (**c**), and weekly (**d**) variations of normalized Au and Hg concentrations in Shanghai.

[Figure]

**Figure S9**. Time series plots of PMF-derived source contributions (in ng m⁻³) for 18 trace elements in PM$_{2.5}$.

[Figure]

**Figure S10.** Spearman correlation matrix of 18 atmospheric elemental species in Shanghai between March 2016 and February 2017. The distribution of each species is shown on the diagonal. Below the diagonal, the bivariate scatter plots with a fitted line are displayed; above the diagonal, the value of the correlation plus the significance level as asterisks. Each significance level is associated to a symbol: *p*-values (0, 0.001, 0.01, 0.05, 0.1, 1) = symbols ("***", "**", "*", "…", ".. ", ". ").

[Figure]

**Figure S11.** Conditional probability function analysis (left) and bivariate polar plots (right) of seasonal concentrations (ng m$^{-3}$) of V (**a**) and Ni (**b**) in Shanghai between 1775 March 2016 and February 2017. The center of each plot (centered at the sampling site) represents a wind speed of zero, which increases radially outward. The concentrations of V and Ni are shown by the color scale.

[Figure]

**Figure S12.** Percentile rose plot of Ag (**a**) and Cd (**b**) concentrations in Shanghai between March 2016 and February 2017. The percentile intervals are shaded and shown by wind direction.

[Figure]

**Figure S13.** Distribution of precipitation events during the full year of measurements.

[Figure]

**Figure S14.** Variation of the mass concentration of each elemental species before, during, and after every precipitation event.

---

## Author Response (AR5)

Dear Prof. Maenhaut,

Thanks again for your careful revision. Below please see our point-by-point reply to your comments.

Comments for the Main text:

*1. My comment about significant figures is still not properly addressed. For example, within Figure 2, for Si, max should be 14700, mean 640, median 360, and 95th percentile 2260. Within Figure 3, for traffic, it should be 680, for shipping 87, and so on. In line 339, it should be "640 ± 1010", in line 340 "410 ± 390", and so on. In line 409, it should be "90%, 50%, 77%, and 63%". In line 427, it should be "640".*

Reply: All the significant figures in the texts, tables, and figures in "Results and discussion" section have been carefully checked and corrected. Here we show the revised versions of figure 2 and figure 3.

[Figure]

date

value

Percent of Total

2. *A few references in the Reference list are not in the appropriate order:*

*"Paatero and Tapper, 1994" should come before "Paatero et al., 2014".*

Reply: Revised accordingly.

Comment for the Supplement:

*Page 7, line 1 above Figure S1: Replace "predetermined" by "selected".*

Reply: Revised accordingly.